# Evolutionary expansion of apical extracellular matrix is required for the elongation of cells in a novel structure

Sarah Jacquelyn Smith[1], Lance A Davidson[2], Mark Rebeiz[1]*

[1]Department of Biological Sciences, University of Pittsburgh, Pittsburgh, United States; [2]Department of Bioengineering, University of Pittsburgh, Pittsburgh, United States

**Abstract** One of the fundamental gaps in our knowledge of how novel anatomical structures evolve is understanding the origins of the morphogenetic processes that form these features. Here, we traced the cellular development of a recently evolved morphological novelty, the posterior lobe of *D. melanogaster*. We found that this genital outgrowth forms through extreme increases in epithelial cell height. By examining the apical extracellular matrix (aECM), we also uncovered a vast matrix associated with the developing genitalia of lobed and non-lobed species. Expression of the aECM protein Dumpy is spatially expanded in lobe-forming species, connecting the posterior lobe to the ancestrally derived aECM network. Further analysis demonstrated that Dumpy attachments are necessary for cell height increases during posterior lobe development. We propose that the aECM presents a rich reservoir for generating morphological novelty and highlights a yet unseen role for aECM in regulating extreme cell height.

## Introduction

Biologists have long been mesmerized by the appearance of morphological novelties, new structures that appear to lack homologs in other species groups (*Moczek, 2008*; *Wagner and Lynch, 2010*). To understand the origins of these novel structures, significant effort has focused on determining how the spatial and temporal patterning of genes were altered during evolution (*Peter and Davidson, 2015*; *Rebeiz et al., 2015*; *Wagner, 2014*). This research has demonstrated that morphological novelties are often associated with the redeployment of pre-existing developmental programs in new locations. However, limited attention has been directed to how novel structures form at the cellular level. Understanding how a structure physically forms is important, as it can help explain which morphogenetic processes might be targeted during evolution. In addition, because most morphological novelties arose in the distant past, it is likely that the causative genetic changes will be obscured by additional changes scattered throughout relevant gene regulatory networks (*Liu et al., 2019*). Hence, understanding the morphogenetic basis of a novelty is critical to identifying the most important aspects of the gene regulatory networks that contributed to its origin.

Most studies of morphogenetic evolution have focused on epithelial structures subject to diversification, illuminating processes that contributed to their modification, as opposed to origination. For example, studies of tooth morphogenesis have elucidated how both internal mechanisms, such as cell shape changes (*Li et al., 2016*), and external forces, such as the pressure from the surrounding jaw (*Renvoisé et al., 2017*) could be contributing factors in their diversification. Furthermore, an examination of the enlarged ovipositor of *Drosophila suzukii* revealed how a 60% increase in length was associated with increases in apical area and anisotropic cellular rearrangement, elucidating how increases in the size of a structure might evolve (*Green et al., 2019*). Dramatic differences in morphogenetic mechanisms have also been observed between distantly related species which

*For correspondence:
rebeiz@pitt.edu

**Competing interests:** The authors declare that no competing interests exist.

contribute to the development of conserved structures, such as eggshell breathing tubes (*Osterfield et al., 2015*) and the migration of sex comb precursors (*Atallah et al., 2009*; *Tanaka et al., 2009*) in *Drosophila.* Overall, these studies have illustrated how evolutionary comparative approaches can reveal morphogenetic processes critical to the sculpting of anatomical structures.

Morphogenesis is the product of both cell intrinsic processes, such as those conferred by the cytoskeleton or cell-cell junctions, and cell extrinsic processes from the environment in which the cell resides. Extracellular mechanics are relatively understudied compared to intracellular mechanics (*Paluch and Heisenberg, 2009*). An important component of the microenvironment of a cell is the extracellular matrix (ECM) which can be subdivided into two populations of ECM, the basal ECM and the apical ECM (aECM) (*Brown, 2011*; *Daley and Yamada, 2013*; *Linde-Medina and Marcucio, 2018*; *Loganathan et al., 2016*). While comparatively understudied, recent work has defined vital roles for aECM in the morphogenesis of various structures in *Drosophila,* such as the wing (*Diaz-de-la-Loza et al., 2018*; *Etournay et al., 2015*; *Ray et al., 2015*), denticles (*Fernandes et al., 2010*), trachea (*Dong et al., 2014*; *Rosa et al., 2018*) and the posterior endoderm (*Bailles et al., 2019*). In addition, roles for the aECM have been found in neuron morphogenesis in *C. elegans* (*Heiman and Shaham, 2009*; *Low et al., 2019*) and during gastrulation in beetles (*Münster et al., 2019*). Despite recent interest in the aECM, its role in the evolution of morphogenetic processes is currently unknown.

Genital traits represent a particularly advantageous system in which to study the morphogenetic basis of novel structures. The study of morphological novelty is often difficult because most structures of interest evolved in the distant past, rendering it difficult to understand the ancestral ground state from which the novelty emerged. Genitalia are noted for their rapid evolution (*Eberhard, 1985*), and thus bear traits among closely-related species that have recently evolved in the context of a tissue that is otherwise minimally altered. For example, the posterior lobe, a recently evolved anatomical structure present on the genitalia of male flies of the *melanogaster* clade (*Kopp and True, 2002*; *Figure 1A*), is a three-dimensional outgrowth that is required for genital coupling (*Frazee and Masly, 2015*; *Jagadeeshan and Singh, 2006*; *LeVasseur-Viens, 2015*). Other than the posterior lobe, the genitalia of lobed and non-lobed species are quite similar in composition, providing an excellent comparative context in which to examine the morphogenesis of the ancestral structure from which the posterior lobe emerged.

Here, we find that the posterior lobe forms through dramatic increases in cell height along the apico-basal axis as it projects out of the plane of its surrounding epithelium. We investigated internal and external factors that might contribute to this cell height increase and discovered a correlation between the aECM protein Dumpy and the height reached by posterior lobe cells. Comparisons to non-lobed species uncovered the presence of a conserved aECM network on the genitalia that has expanded to cells that form the posterior lobe. Overall, our work demonstrates how the formation of a morphological novelty is associated with changes in the aECM, which integrates posterior lobe cells into a larger pre-existing aECM network necessary for its formation.

## Results

### The posterior lobe grows from the lateral plate epithelium

The male genitalia of *Drosophila* is a bilaterally symmetrical anatomical structure which forms from the genital disc during pupal development. A recent consortium detailed the adult anatomy of *Drosophila melanogaster* males, establishing naming conventions and boundaries for adult structures (*Rice et al., 2019*). The posterior lobe (also known as the epandrial posterior lobe) protrudes from a structure called the lateral plate (also known as the epandrial ventral lobe) (*Figure 1A,D*; *Figure 1— video 1*). In *D. melanogaster*, prior to posterior lobe formation, the lateral plate is fully fused to a neighboring structure called the clasper (also known as the surstylus) (*Figure 1B*; *Glassford et al., 2015*). The lateral plate begins to separate from the clasper around 32 hr after pupal formation (APF) in *D. melanogaster* (*Figure 1—figure supplement 1*). Approximately 4 hr later, the posterior lobe begins to project from the plane of the lateral plate and achieves its final shape by 52 hr APF (*Figure 1D*; *Figure 1—figure supplement 1*). During posterior lobe development, cleavage of the lateral plate from the clasper continues, separating both tissues (*Figure 1D*; *Figure 1—figure*

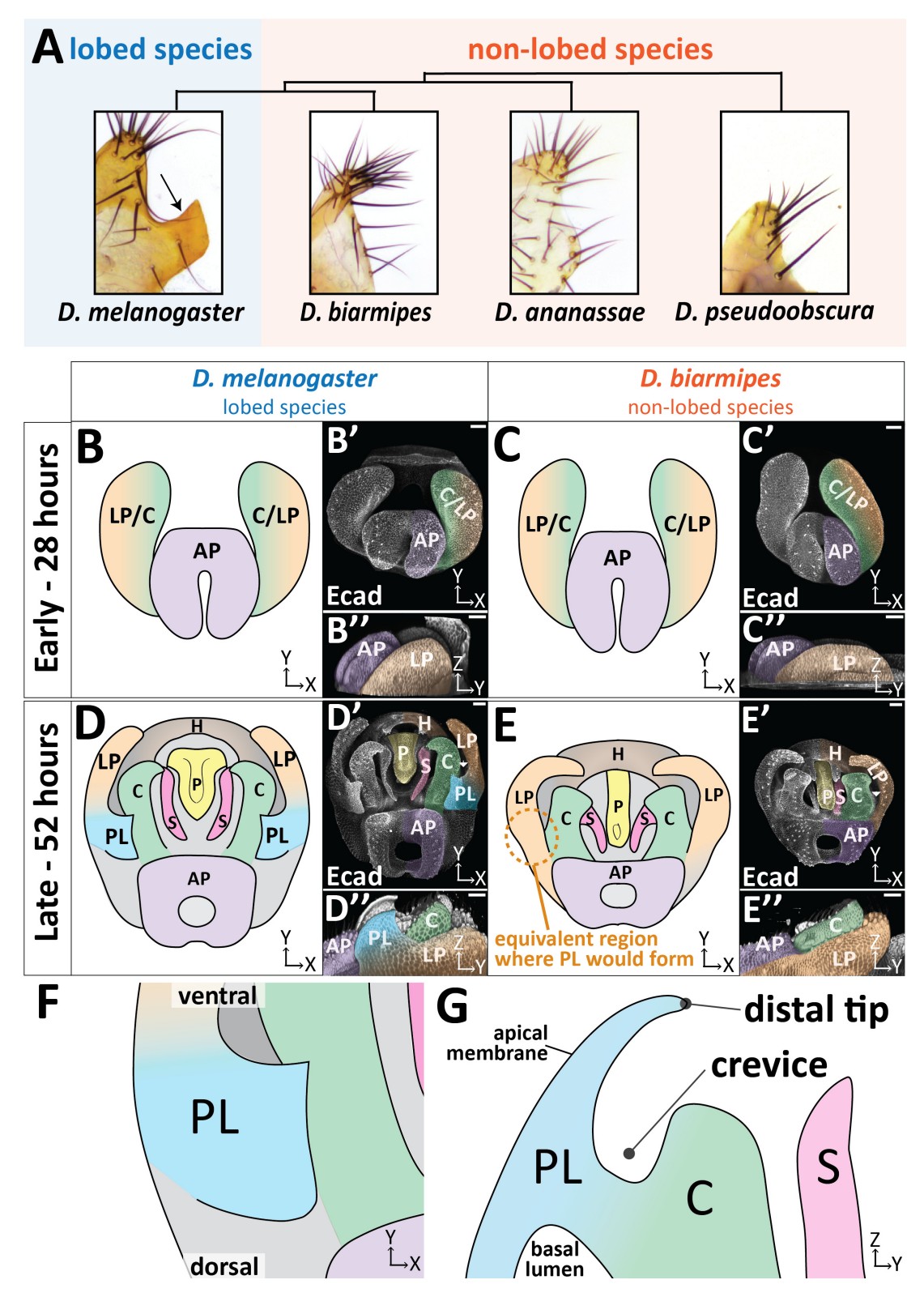

**Figure 1.** The posterior lobe protrudes from the lateral plate. (A) Phylogenetic tree with bright-field images of adult lateral plate cuticles from which the posterior lobe projects (arrow). (B–E) Illustration, (B'–E') maximum projection, and (B''–E'') three-dimensional projection labeled with E-cadherin (Ecad) of early (28 hr APF) and late (52 hr APF) developing genitalia showing the posterior lobe projecting form the lateral plate of *D. melanogaster* (D''), but absent in *D. biarmipes* (E''). Relevant structures are labeled: posterior lobe (PL), lateral plate (LP), clasper (C), sheath (S), phallus (P), anal plate (AP), and
*Figure 1 continued on next page*

*Figure 1 continued*

hypandrium (H). All max projections are oriented with ventral side towards to the top and dorsal sides towards the bottom. (**F**) Zoomed in illustration of posterior lobe and (**G**) a cross-sectional/lateral view of the posterior lobe. The highest point of the lobe is the distal tip and the invagination between the lobe and the clasper is termed the crevice (**G**). Scale bar, 20 μm.

The online version of this article includes the following video and figure supplement(s) for figure 1:

**Figure supplement 1.** Developmental timing of lobed vs non-lobed genitalia.

**Figure 1—video 1.** The posterior lobe protrudes from the lateral plate.

https://elifesciences.org/articles/55965#fig1video1

---

*supplement 1*). Full separation of the lateral plate and clasper stops slightly ventral to the posterior lobe (*Figure 1—figure supplement 1*). By contrast, in the non-lobed species *D. biarmipes,* the apical surface of the lateral plate remains flat throughout development, not forming a posterior lobe (*Figure 1C,E*). However, all other morphogenetic events are very similar in *D. biarmipes*, forming on a schedule that is approximately 4 hr behind *D. melanogaster* (*Figure 1—figure supplement 1*).

## The posterior lobe forms through increases in cell height

To investigate which cellular behaviors are unique to lobed species, we examined how the posterior lobe grows from the lateral plate. First, we monitored cell proliferation, which commonly contributes to morphogenesis through patterned and/or oriented cell division (*Heisenberg and Bellaïche, 2013*). Staining for phospho-histone H3, which marks actively dividing cells, revealed widespread cell proliferation throughout the entire genital epithelium during early developmental stages prior to posterior lobe morphogenesis (*Figure 2—figure supplement 1*). However, proliferation declines tissue-wide and all cell proliferation is essentially absent during posterior lobe development (*Figure 2—figure supplement 1*). Similar dynamics in proliferation are also observed in non-lobed species (*Figure 2—figure supplement 1*), suggesting that proliferation is not a major contributor to the morphogenesis of the posterior lobe.

Next we tested the possibility that oriented cell intercalation could contribute to posterior lobe morphogenesis. Such processes may play a role in tissue elongation (*Guirao and Bellaïche, 2017*; *Tada and Heisenberg, 2012*; *Walck-Shannon and Hardin, 2014*). To test this, we performed manual cell tracking of time-lapse movies of posterior lobe development in a GFP-tagged armadillo line (*Huang et al., 2012*) which labels apical cell junctions. Initial observations of the outer face of the posterior lobe revealed few cell rearrangement events. When cell rearrangements did occur, it was usually in response to a cell being removed from the apical surface (*Figure 2—video 1*). Due to the limited number of cell rearrangement events observed during posterior lobe morphogenesis, cell intercalation did not appear to be a major driver of posterior lobe morphogenesis, causing us to instead examine changes in cell shape.

Changes to cell shape are quite common during tissue morphogenesis, as classically illustrated by the process of apical constriction which deforms tissues during numerous developmental processes (*Lecuit and Lenne, 2007*; *Martin and Goldstein, 2014*). To examine cell shape, we utilized the Raeppli system to label individual cells with a fluorescent marker (mTFP1) (*Kanca et al., 2014*). We observed that cells within the posterior lobe are tall and thin, spanning from the basal to the apical surface of the epithelium (*Figure 2A*). Because cells span the full thickness of this tissue, we can approximate the height of the tallest cells in the posterior lobe by measuring tissue thickness in samples double-stained for E-cadherin and Fasciclin III, to label apical cell junctions and lateral membranes respectively. For these measurements, we used the lateral plate as an in-sample comparison, since it represents the tissue from which the posterior lobe protrudes and should differ from the lobe in morphogenetic processes. We observed a pronounced increase in thickness of the posterior lobe compared to the lateral plate (*Figure 2B–C,F*; *Figure 2—figure supplement 2*). The posterior lobe more than doubles in thickness with an average increase of 145.3% (+ 47.5 μm), while the lateral plate only increases by 22.6% (+ 7.9 μm) overall. Interestingly, this is a dynamic process during development. During the first 12 hr of posterior lobe development, the lateral plate thickness decreases by 5.1 μm, but the posterior lobe increases in thickness by 16.5 μm on average (*Figure 2F*). By contrast, during the last 4 hr of development, rapid increases in thickness occur in both the posterior lobe and lateral plate, which increase on average by 31.0 μm and 14.6 μm

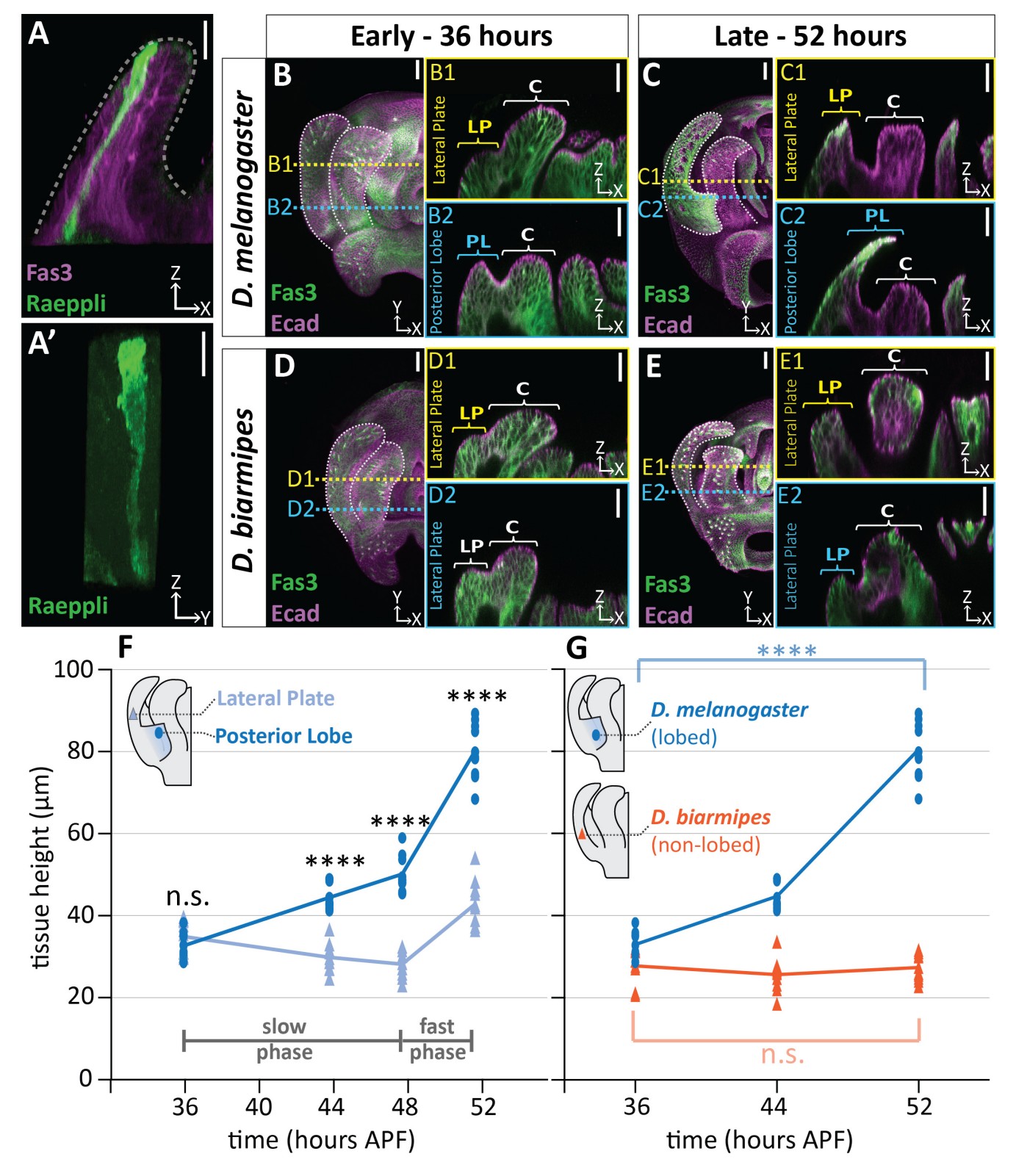

**Figure 2.** Posterior lobe cells increase in height to project out from the lateral plate. (A) A single cell in the posterior lobe labeled with Raeppli-mTFP1 (green) spans the height of the tissue labeled with lateral membrane marker Fasciclin III (Fas3; magenta). Apical side of posterior lobe identified with dotted line. Sample is 44 hr after pupal formation (APF), but was heat shocked for 1 hr at 24 hr APF causing it to develop faster and more closely resembles a 48 hr APF sample. Scale bar, 10 μm. n = 10 cells (B–E) Maximum projections of early (36 hr APF) and late (52 hr APF) genital samples

*Figure 2 continued on next page*

*Figure 2 continued*

labeled with Fas3 (lateral membranes, green) and E-Cadherin (Ecad; apical membranes; magenta). Location of respective cross sections indicated in yellow for lateral plate (B1–E1) and blue for posterior lobe (*D. melanogaster*) (B2–C2) or equivalent location in non-lobed species (*D. biarmipes*) (D2–E2). Scale bar, 20 µm. Relevant structures are labeled: posterior lobe (PL), lateral plate (LP), clasper (C). (F) Quantification of tissue thickness of the lateral plate (light blue) and posterior lobe (dark blue). Illustration represents approximate location of cross-section that was used for tissue height measurement. Individual data points are presented; n = 10 per each time point. (G) Quantification of tissue thickness of the posterior lobe in *D. melanogaster* (dark blue) and equivalent location in non-lobed species *D. biarmipes* (orange). Illustration represents approximate location of cross-section that was used for tissue thickness measurement. Individual data points are presented; n ≥ 9 per each time point. Statistical significance is indicated (unpaired t-test; ****p≤0.0001; n.s. = not significant p≥0.05). *D. melanogaster* tissue height measures in (G) are replotted from (F) to facilitate direct comparisons with *D. biarmipes*.

The online version of this article includes the following video, source data, and figure supplement(s) for figure 2:

**Source data 1.** Individual measurements of tissue thickness over time.
**Figure supplement 1.** Cell division dynamics do not differ between lobed and non-lobed species.
**Figure supplement 2.** Extended time course of tissue thickness in lobed and non-lobed species.
**Figure 2—video 1.** Cell rearrangement during posterior lobe development.
https://elifesciences.org/articles/55965#fig2video1

respectively (*Figure 2F*). These observations reveal a slow phase of cell height increase during the first 12 hr of posterior lobe development, and a fast phase during the last four hours of posterior lobe development. In contrast, when non-lobed species are examined, no thickness changes are observed in the location where a posterior lobe would form, indicating that this increase in tissue thickness is unique to the posterior lobe (*Figure 2B–E,G*; *Figure 2—figure supplement 2*). Together, these data suggest that the cells of the posterior lobe undergo an extreme shape change to increase in length along their apico-basal axes, driving the projection of this structure out of the plane of the lateral plate.

## Cytoskeletal components increase in concentration in posterior lobe cells

Apico-basal cell elongation appears to be a major contributor to posterior lobe formation. To understand potential internal forces contributing to this cell shape change, we examined the organization of cytoskeletal components. As expected for a polarized epithelium, phalloidin staining of F-actin strongly localized to the apical cortex overlapping with E-cadherin throughout the entire genitalia (*Figure 3A*). In addition, F-actin is also concentrated along the apico-basal axis of posterior lobe cells (*Figure 3A*). This lateral F-actin localization was unique to the posterior lobe, as it is less intense in neighboring structures, such as the lateral plate, clasper, and sheath (also known as the aedeagal sheath *Rice et al., 2019*), as well as in non-lobed species (*Figure 3A*; *Figure 3—figure supplement 1*). Next we evaluated microtubules by examining two post-translational modifications of α-tubulin: acetylation of lysine40, a stabilizing modification (*Roll-Mecak, 2019*; *Xu et al., 2017b*), and C-terminal tyrosination, which has been associated with rapid microtubule turnover (*Roll-Mecak, 2019*; *Webster et al., 1987*). In the posterior lobe, acetylated tubulin levels are highest at the distal tip of the posterior lobe and weaken towards the basal side of the lobe epithelium (*Figure 3B–C*). Compared to other structures in the genitalia, acetylated tubulin is greatly increased specifically in the posterior lobe (*Figure 3B–C*). By contrast, the levels of acetylated tubulin in non-lobed species are similar throughout the genitalia (*Figure 3—figure supplement 1*). We found tyrosinated tubulin has a more consistent signal along the entire apico-basal axis in the posterior lobe (*Figure 3B&D*). The amount of tyrosinated tubulin in posterior lobe cells is increased compared to neighboring structures but is weaker relative to the observed differences in acetylated tubulin. In non-lobed species, the levels of tyrosinated tubulin are consistent across the entire genitalia (*Figure 3—figure supplement 1*). Collectively, these results suggest that changes in assembly and/or dynamics of both F-actin and microtubule cytoskeletal networks could be contributing factors in changing the shape of posterior lobe cells to increase its height along the apico-basal axis.

## An apical extracellular matrix associates with posterior lobe cells

In addition to investigating potential cell autonomous mechanisms leading to increases in tissue thickness, we also sought to identify possible sources of external mechanical processes which could

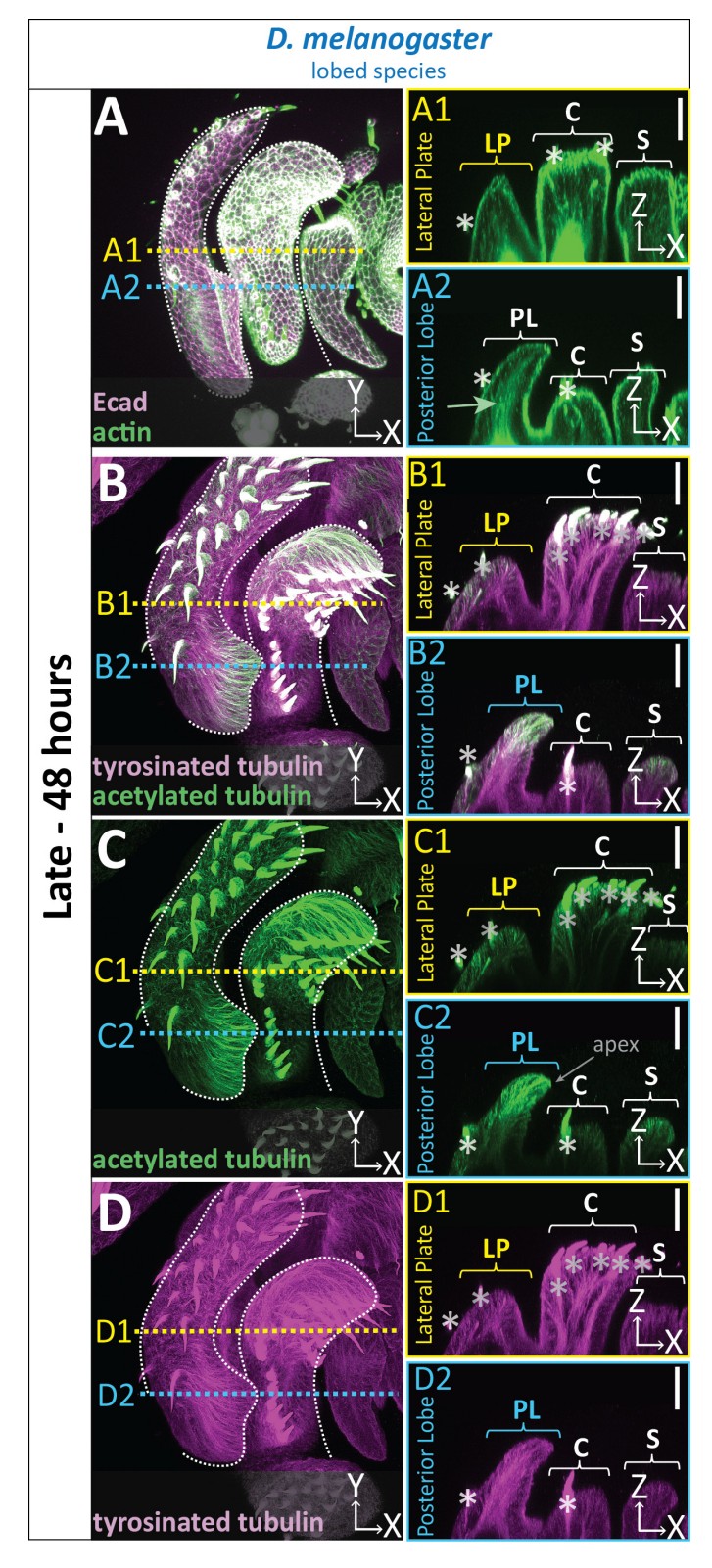

**Figure 3.** Cytoskeletal components are increased in posterior lobe cells. (A–D) Maximum projection, and respective cross-sections of late (48 hr APF) genital samples of the lobed species *D. melanogaster* labeled with F-actin/phalloidin (actin; green) and E-cadherin (Ecad; magenta) (A), acetylated tubulin (green) (B,C), and tyrosinated tubulin (magenta) (B,D). Location of respective cross sections indicated in yellow for lateral plate (A1–

*Figure 3 continued on next page*

*Figure 3 continued*

D1) and blue for posterior lobe (**A2–D2**). Cross-sections are maximum projections of a restricted 5.434 µm thick section to provide a complete view of cytoskeletal components along the apico-basal axis. All cross-sections are oriented with apical side at the top and basal side at the bottom. Asterisk identifies bristles which have high levels of F-actin and tubulin. Bright basal signal in A1 and A2 are fat bodies. Bottom layers were removed in panel A to remove fat body signal which overwhelmed other details. (**B–D2**) Panels C and D show separate channels of panel B. Relevant structures labeled: Posterior lobe (PL), lateral plate (LP), clasper (C), and sheath (S). Scale bar, 20 µm. n $\geq$ 3 per experiment.

The online version of this article includes the following figure supplement(s) for figure 3:

**Figure supplement 1.** Uniform level of cytoskeletal components in non-lobed species.

play a role in posterior lobe morphogenesis. Extrinsic roles for both the basal and apical extracellular matrix have been established in pupal wing morphogenesis of *D. melanogaster* (***Diaz-de-la-Loza et al., 2018***; ***Etournay et al., 2015***; ***Ray et al., 2015***). We first attempted to characterize the basal ECM by analyzing a GFP-tagged version of Collagen IV (Viking:GFP) (***Morin et al., 2001***). We observed that Viking:GFP, while present at very early stages of genital morphogenesis, is only weakly detected across the entire genitalia as the posterior lobe forms (***Figure 4—figure supplement 1***), suggesting that minimal basal ECM is present at this time point. To further test for the presence of basal ECM, we examined another basal ECM component, Perlecan (Perlecan:GFP) (***Morin et al., 2001***), and also observed weak signal (***Figure 4—figure supplement 1***). Together, these data suggest that the basal ECM is globally reduced in the genitalia during early pupal development, and it thus is unlikely to have specific effects on posterior lobe morphogenesis.

We next sought to determine if an apical ECM (aECM) is associated with the posterior lobe. *Dumpy* encodes a gigantic (2.5 MDa) zona pellucida domain-containing glycoprotein and is a major component of the aECM in *Drosophila* (***Wilkin et al., 2000***). We examined a line in which Dumpy is endogenously tagged with Yellow Fluorescent Protein (Dumpy:YFP) (***Lowe et al., 2014***; ***Lye et al., 2014***) and found that Dumpy:YFP forms a complex three-dimensional network over the pupal genitalia and is closely associated with cells of the posterior lobe (***Figure 4***; ***Figure 4—video 1***). The intricate complex morphology of this aECM network is hard to fully appreciate in flattened images due to its three-dimensional shape and spatially varying levels of Dumpy:YFP, making it difficult to see weaker populations of Dumpy without over-saturating more concentrated deposits, and is better viewed in three dimensions (***Figure 4—video 1***). Remarkably, at certain points in the genitalia, this aECM network of Dumpy can extend up to a mean maximal height of 39.4 µm above the cells, which is taller than the thickness of posterior lobe cells at the beginning of development (***Figure 4—figure supplement 2***), demonstrating how extensive the genital aECM network is.

In late pupal wing development, Dumpy anchors the wing to the surrounding cuticle, holding this tissue in place, which is important to properly shape the wing (***Etournay et al., 2015***; ***Ray et al., 2015***). This same mechanism has been hypothesized to also occur in the leg and antennae (***Ray et al., 2015***), however, in the posterior lobe we do not find discrete anchorage points to the cuticle. Instead, we observed a large bundle of Dumpy emanating from the anal plate (also known as the cercus ***Rice et al., 2019***) and connecting with the pupal cuticle that encases the entire pupa (***Figure 4—figure supplement 3***, ***Figure 4—video 2***; ***Bainbridge and Bownes, 1981***). This bundle does not come in direct contact with posterior lobe associated Dumpy or other nearby structures such as the lateral plate, clasper, sheath, or phallus, suggesting that if Dumpy is contributing to posterior lobe evolution and morphogenesis, this likely occurs through a mechanism which does not depend on a direct mechanical linkage with the overlying pupal cuticle.

To investigate the role that Dumpy may play in posterior lobe morphogenesis, we examined its localization throughout development. Prior to posterior lobe development, future cells of the lobe lack apically localized Dumpy, and yet an intricate network associated with the presumptive clasper is observed (***Figure 4A***). However, from the early stages of posterior lobe development, as it first protrudes from the lateral plate, we observe large deposits of Dumpy associated with future lobe cells (***Figure 4B***). These deposits persist throughout its development (***Figure 4***), becoming more restricted to the distal tip of the posterior lobe towards the end of its formation (***Figure 4 D2***). Across all timepoints, the posterior lobe associated Dumpy population is connected to the complex network of Dumpy attached to more medial structures such as the sheath and phallus (***Figure 4 A2–***

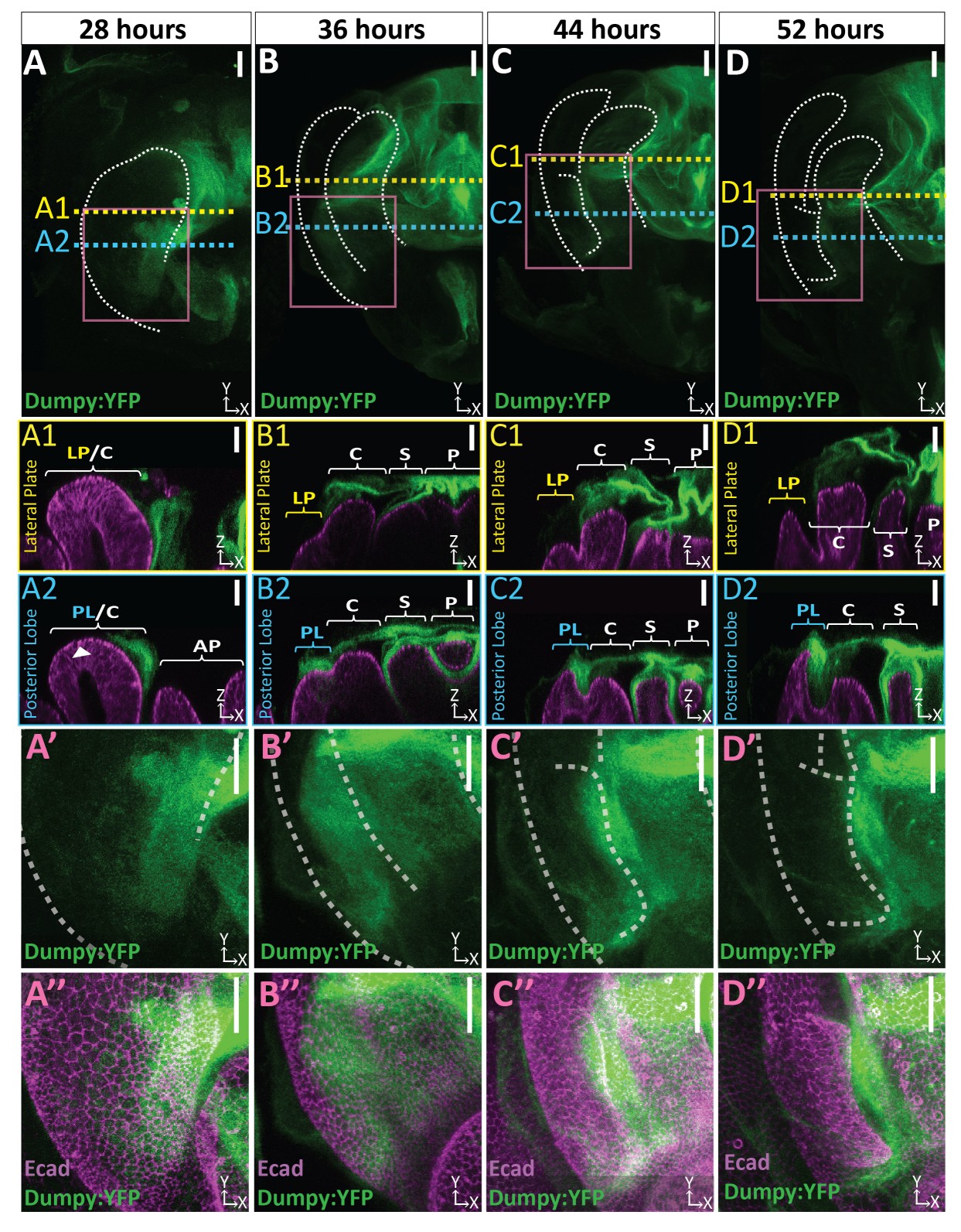

**Figure 4.** Dumpy deposition is correlated with posterior lobe development. (**A–D**) Maximum projection and (**A'–B''**) respective zoom, indicated with pink box, labeled with Dumpy:YFP (green) and E-cadherin (Ecad; magenta) for each time point. Location of respective cross sections indicated in yellow for lateral plate (**A1–D1**) and blue for posterior lobe (**A2–D2**). Arrowhead in (**A2**) indicates future posterior lobe cells. Cross-sections are oriented with

*Figure 4 continued on next page*

*Figure 4 continued*

apical side at the top and basal side at the bottom. Relevant structures labeled: Posterior lobe (PL), lateral plate (LP), clasper (C), sheath (S), and phallus (P). Scale bar, 20 μm. n ≥ 4 per experiment. Images were independently brightened to show relevant structures.

The online version of this article includes the following video and figure supplement(s) for figure 4:

**Figure supplement 1.** Limited basal ECM present during posterior lobe morphogenesis.
**Figure supplement 2.** Dumpy extends above the apical surface of the phallus.
**Figure supplement 3.** A bundle of Dumpy connects the genitalia to the pupal cuticle membrane that encases the developing pupa.
**Figure supplement 4.** Weak aECM connections extend to the lateral plate.
**Figure 4—video 1.** Three-dimensional structure of Dumpy on developing genitalia.
https://elifesciences.org/articles/55965#fig4video1
**Figure 4—video 2.** A bundle of Dumpy connects the genitalia to the surrounding cuticle.
https://elifesciences.org/articles/55965#fig4video2

---

*D2*), indicating that the posterior lobe is interconnected via the aECM with nearby structures (*Figure 4*). In contrast to the posterior lobe, the lateral plate has minimal Dumpy associated with it (*Figure 4 A1–D1*). Only when we oversaturate the Dumpy:YFP signal can we observe a weak population of Dumpy associated with the lateral plate (*Figure 4—figure supplement 4*). Together, this indicates that the cells of the posterior lobe and the lateral plate substantially differ in the levels of associated Dumpy, suggesting a potential role in the morphogenesis of the posterior lobe.

## Expansion of *dumpy* expression is correlated with the evolution of the posterior lobe

The association of the posterior lobe with Dumpy suggests that changes in its expression pattern may have occurred during evolution of the lobe. To test if posterior lobe-associated Dumpy is a unique feature of species which produce a posterior lobe, we compared the spatial distribution of its mRNA in *D. melanogaster* with *D. biarmipes,* a species which lacks this structure. Early in pupal genital development at 32 hr APF, we observe very similar expression patterns of *dumpy* between *D. melanogaster* and *D. biarmipes* with expression at the base of the presumptive lateral plate-clasper (*Figure 5A–B*, *Figure 5—figure supplement 1*). From 36 to 40 hr APF, when the posterior lobe begins to develop, this pattern becomes restricted to a small region at the base of the lateral plate and clasper, near the anal plate in *D. biarmipes*, but is expanded in *D. melanogaster* (*Figure 5A–B*, *Figure 5—figure supplement 1*). By 44 hr APF, expression of *dumpy* is reduced in the posterior lobe, as well as in non-lobed species, with strongest expression associated with the clasper in *D. biarmipes* (*Figure 5A–B*, *Figure 5—figure supplement 1*). Overall, these results indicate that expression of *dumpy* is expanded in a lobed species and correlates with the developmental timing of the posterior lobe's formation.

Although, it appears that the expression of *dumpy* has expanded in *D. melanogaster*, Dumpy is an extracellular protein, and cells expressing its mRNA may not correlate with its ultimate protein abundance or localization. Since an antibody for Dumpy is not available, we adapted lectin staining protocols which can detect glycosylated proteins like Dumpy in order to compare the distribution of aECM in species which lack posterior lobes. We found that fluorescein conjugated *Vicia villosa* lectin (VVA), which labels *N*-acetylgalactosamine (*Tian and Ten Hagen, 2007*), roughly mirrors the complex three-dimensional shape of the Dumpy aECM network covering the center of the genitalia and strongly associates with the posterior lobe (*Figure 5C*, *Figure 5—figure supplement 2*). When we examined VVA in the non-lobed species *D. biarmipes*, we observed strong VVA signal over the center of the genitalia with weak connections to the tip of the lateral plate, similar to what we observe in *D. melanogaster* (*Figure 5C–D*). In contrast, where the presumptive posterior lobe would form, we only found a weak strand-like structure emanating from the clasper and connecting to the crevice between the lateral plate and clasper (*Figure 5D1*). These results correlate with our *dumpy* in situ hybridization results, where we observe high expression at the center of the genitalia and weak expression at the base between the clasper and lateral plate in *D. biarmipes*, which may be responsible for forming the weak aECM connection from the clasper to the clasper/lateral plate crevice. Further, we observed similar VVA patterns in non-lobed species *D. ananassae* (*Figure 5—figure supplement 3*). In addition, in the lobed species, *D. sechellia*, we note a similar accumulation of VVA associated with the developing lobe as seen in *D. melanogaster* (*Figure 5—figure supplement 3*),

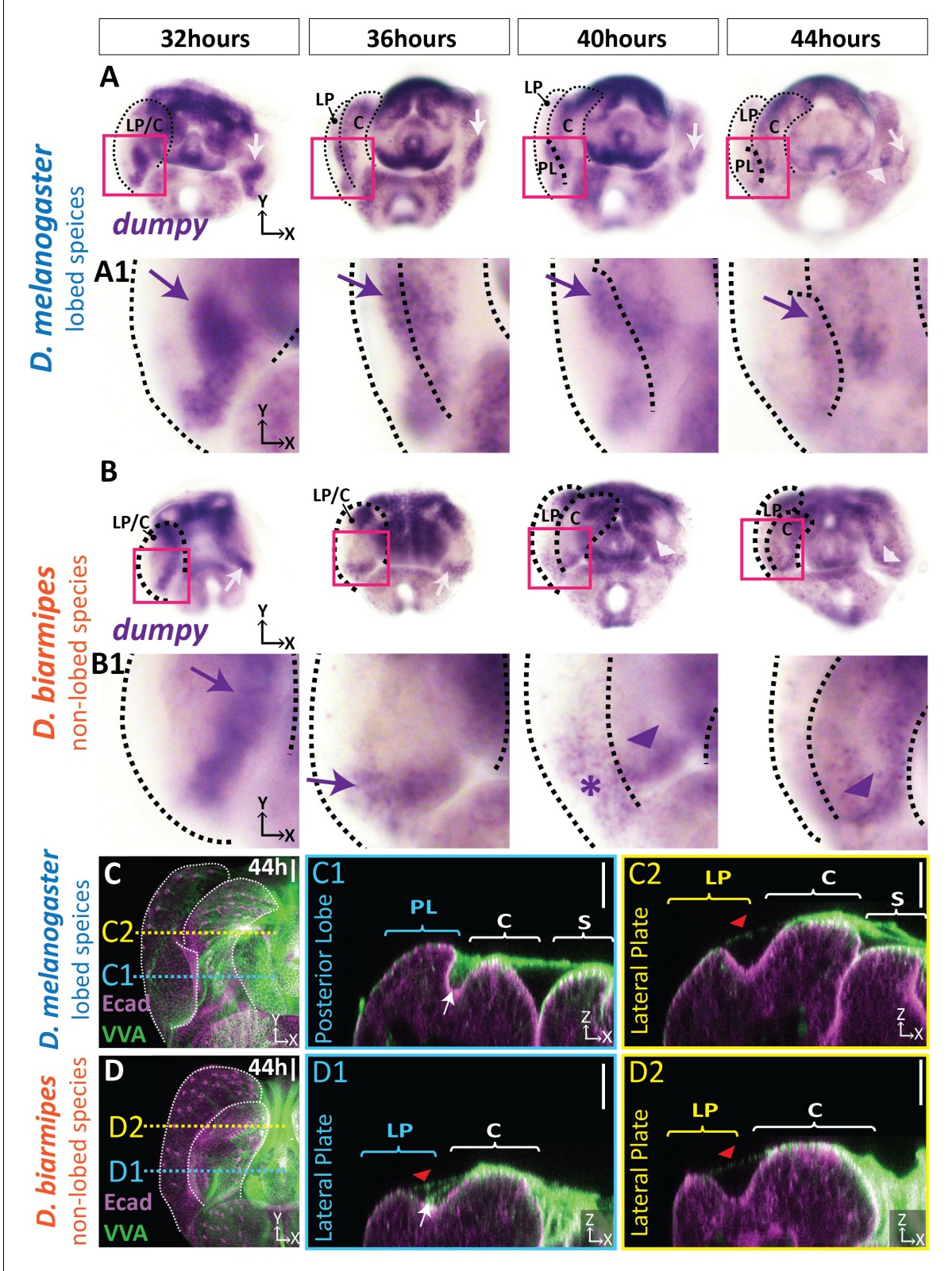

**Figure 5.** aECM is spatially expanded in lobed species compared to non-lobed species. (A–B) in situ hybridization for *dumpy* mRNA in the lobed species *D. melanogaster* (A) and the non-lobed species *D. biarmipes* (B). Pink box outlines location of zoomed in images presented in A1 and B1. Relevant expression highlighted with arrow (purple/white) for strong expression, asterisk for weak expression, and arrowhead for clasper-specific expression. Expression observed in *D. melanogaster* at 44 hr APF is not present in all samples (see *Figure 5—figure supplement 1*). (C–D) aECM is

*Figure 5 continued on next page*

*Figure 5 continued*
labeled with *Vicia villosa* lectin (VVA; green) and apical membrane labeled with E-cadherin (Ecad; magenta) at 44 hr APF in *D. melanogaster* (**C**) and *D. biarmipes* (**D**). Location of respective cross sections indicated in yellow for lateral plate (**C2–D2**) and blue for posterior lobe in *D. melanogaster* (**C1**) and corresponding position in *D. biarmipes* (**D1**). All cross-sections are oriented with apical side at the top and basal side at the bottom. White arrows highlight the crevice localization between the lateral plate and clasper, which the aECM fills in *D. melanogaster* (**C1**), but only a weakly stained strand-like structure of aECM appears in *D. biarmipes* (**D1**). Tendrils of aECM can also be observed connecting to the lateral plate in both species (red arrowheads). Relevant structures labeled: Posterior lobe (PL), lateral plate (LP), clasper (C), sheath (S), and phallus (P). Scale bar, 20 μm. n = at least five per experiment.
The online version of this article includes the following figure supplement(s) for figure 5:

**Figure supplement 1.** aECM spatially expanded in lobed species compared to non-lobed species.
**Figure supplement 2.** VVA staining mimics Dumpy:YFP localization.
**Figure supplement 3.** aECM is expanded in the lobed *D. sechellia* but not in the non-lobed species *D. ananassae*.

suggesting that this mechanism is common to other lobe-bearing species. Collectively, these data suggest that an ancestral aECM network exists on the developing genitalia, associated with the central genital structures, including the phallus, sheath, and clasper, and extends weak connections to the crevice between the lateral plate and clasper. During the course of evolution, expression of *dumpy* expanded to cells of the posterior lobe, creating prominent associations of the lobe cells with this ancestral aECM network.

## Dumpy is required for proper posterior lobe formation

Thus far, we observed a strong association of the aECM with cells that form the posterior lobe, an attribute which is much less pronounced in non-lobed species. To determine if Dumpy plays a role in posterior lobe formation, we next employed transgenic RNAi to knock down its expression. Previous studies of *dumpy* characterized a VDRC RNAi line that is effective at reducing its activity (*Ray et al., 2015*). We used a driver from the *Pox neuro* gene (*Boll and Noll, 2002*) to target the RNAi to posterior lobe cells. This resulted in a drastic decrease in the size of the posterior lobe and also caused changes to its shape (*Figure 6*). In *dumpy* knockdown, we observe a variable phenotype between the left and right posterior lobes, even within a single individual (*Figure 6A*; *Figure 6—figure supplement 1*). Knockdown was completed at both 25℃ and 29℃, as higher temperatures increase the efficacy of the Gal4/UAS system (*Duffy, 2002*). At these higher temperatures, the *dumpy* knockdown phenotype trended towards more severe defects (*Figure 6B*). Together, these results suggest that posterior lobe development is sensitive to levels of *dumpy*, and that *dumpy* plays a vital role in shaping the posterior lobe.

## Correlation of Dumpy deposition and cell height in the posterior lobe

We next sought to determine when during development *dumpy* knockdown influences the morphogenetic progression of the posterior lobe. This was important because the posterior lobe emerges over 16 hr of development (*Figure 1F*), after which cells of the genital epithelium secrete a rigid cuticle, and any of these phases could represent a critical Dumpy-dependent stage of development. We found that *dumpy* knockdown individuals manifest phenotypes by the midpoint of posterior lobe development (*Figure 7A*) and continue to show abnormal lobe development through the end of its formation (*Figure 7B*). Defects in posterior lobe development may occur earlier, but the exact ventral and dorsal boundaries of the posterior lobe are difficult to define for quantification purposes at early time points. Interestingly, while defects in cell height are observed on the dorsal side, we do not see differences in cell height in the ventral cells of the lobe (*Figure 7A–B*). This correlates with the phenotypes of adult *dumpy* knockdown individuals, which usually display a ventral tip of normal height with defects observed towards the dorsal side (*Figure 6A*).

Why would dorsal cells of the posterior lobe be more affected in a *dumpy* knockdown? We hypothesized that our posterior lobe specific driver is not strong enough to remove all deposits of Dumpy associated with the posterior lobe. To better understand this, we examined Dumpy:YFP localization in the *dumpy* knockdown background, in which both *dumpy* and *dumpy:yfp* should be targeted by the RNAi treatment. In the *dumpy* RNAi background, we observed an association of Dumpy:YFP with the tallest cells on the ventral side of the posterior lobe in both mid (*Figure 7D* n = 5/5 samples) and late (*Figure 7F* n = 4/5 samples) stages of development compared to control

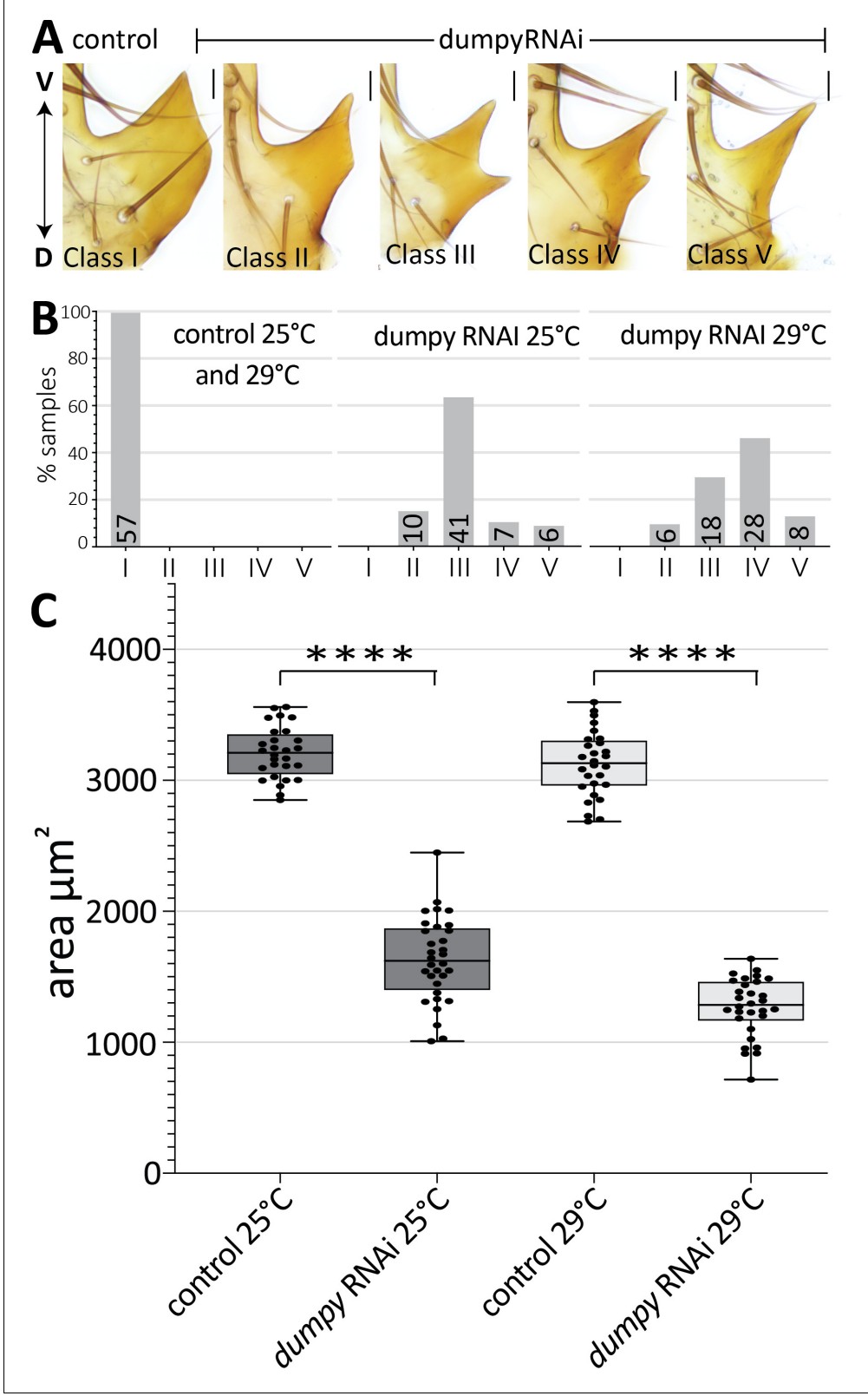

**Figure 6.** Dumpy is required for proper posterior lobe formation. (**A**) Range of adult posterior lobe phenotypes produced by control (*mCherry* RNAi) and *dumpy* RNAi animals. Phenotypic classes defined from wild type (I) to most severe (V). Scale bar, 20 μm. (**B**) Percentage of posterior lobes in each class for control, *dumpy* RNAi at 25°C, and *dumpy* RNAi at 29°C. (**C**) Quantification of area of adult posterior lobes of *mCherry* RNAi (control) and *dumpy* RNAi at 25°C and 29°C. Statistical significance is indicated (unpaired t-test; ****p≤0.0001).

*Figure 6 continued on next page*

*Figure 6 continued*

The online version of this article includes the following source data and figure supplement(s) for figure 6:

**Source data 1.** Individual measurements of *dumpy*-RNAi adult cuticle phenotypes.

**Figure supplement 1.** Increased left-right variability of posterior lobe phenotype upon *dumpy* knockdown.

animals. In contrast, no Dumpy was observed in contact with the shorter cells on the dorsal side (*Figure 7D* and F n = 5/5 samples). In addition, it should be noted that one of our late samples lacks a Dumpy connection to the ventral cells, correlating with our observation that not all adult samples are fully extended on the ventral side (*Figure 7—figure supplement 1*). Furthermore, we also observed highly variable strands of Dumpy:YFP in the middle of the lobe (between the ventral and dorsal sides) (*Figure 7—figure supplement 2*). These strands visually resembled the weak strands of VVA observed in *D. biarmipes* (*Figure 5D*), in that they emanate from the clasper and connect to the crevice between the posterior lobe and clasper, indicating that this 'strand' like connection in the crevice alone is not able to elongate cells.

To further confirm the effect of aECM in the *dumpy* knockdown background, we examined VVA during mid stages of development when defects in posterior lobe formation are evident. We observed a single thin strand of VVA at the ventral tip (*Figure 7—figure supplement 3* n = 3/3) in *dumpy* knockdown individuals. In addition, no VVA was observed at the dorsal side of the posterior lobe (*Figure 7—figure supplement 3* n = 2/3), with exception of one sample where a small strand of VVA was observed (*Figure 7—figure supplement 3C* n = 1/3). This small tether correlates with dorsal spikes observed in some adult individuals (*Figure 6*). Overall, these data demonstrate that the most pronounced phenotypic defects manifest in regions with the strongest reduction in Dumpy deposition, implying that Dumpy's presence is required for posterior lobe cells to elongate and project from the lateral plate.

## Discussion

Here, we investigated how a morphological novelty forms at the cellular level, and in doing so, revealed distinctive cell and aECM interactions underlying its development and evolution. We identified how an extreme change in the shape of cells in the developing posterior lobe accounts for its novel morphology. While intrinsic cytoskeletal components appear to contribute to this process, our results indicate a critical role played by a vast extrinsic network of ECM on the apical side of the epithelium. It was unexpected that such an elaborate supercellular matrix structure would participate in the evolution of a novel morphological structure. Below, we consider the potential roles played by the aECM in posterior lobe development and diversification, and discuss how studies of morphogenesis can illuminate the simple origins of structures that might otherwise seem impossibly complex to evolve.

### Mechanisms for aECM-mediated control of cell height in the posterior lobe

Our work demonstrates an important role for the aECM protein Dumpy in the development of the posterior lobe, as exhibited by the dramatic phenotypes in the *dumpy* RNAi background and the strong association of Dumpy:YFP with only the tallest cells in these experiments. The presence of Dumpy:YFP at the ventral tip of the defective posterior lobe in *dumpy* RNAi individuals indicated that our driver does not supply sufficient RNAi in ventral tip cells to remove all of the Dumpy-YFP, though a strong reduction in florescence is observed at the ventral tip. We hypothesize that this ventral tip would not fully form if Dumpy is completely removed, however, we currently do not have a driver that can test this prediction.

Overall, our data are consistent with three possible mechanisms that could allow Dumpy to regulate cell height. First, Dumpy could serve as a structural support while autonomous cell mechanical processes drive apico-basal elongation, such as by altering the cytoskeleton, which is observed in posterior lobe cells. Second, the cells of the posterior lobe could be pulled mechanically through their connection to the Dumpy aECM. This process could operate passively, deforming cells of the lobe, but could also drive changes in the cytoskeleton in response to external tensions. Finally, the

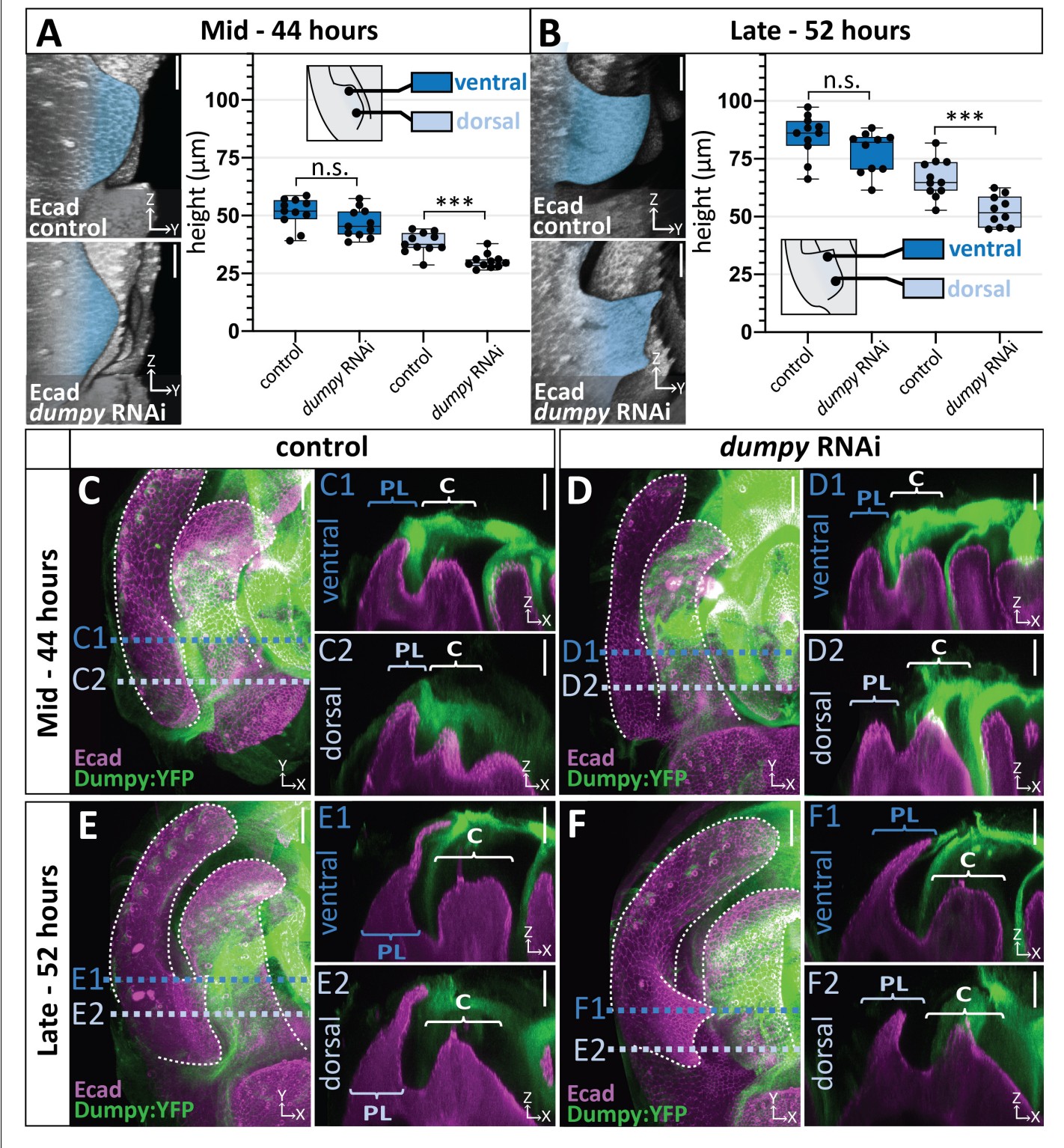

**Figure 7.** Correlation between the deposition of Dumpy and knockdown phenotype. (A–B) Comparison of *mCherry* RNAi (control) and *dumpy* RNAi at 44 hr APF (A) and 52 hr APF (B). Images are rotated in 3D to visualize the full shape of the posterior lobe labeled with E-cadherin (Ecad). Quantification of tissue height at the ventral tip (dark blue) and dorsal base (light blue) of the posterior lobe. Cartoon represents relative locations of cross-sections used for tissue thickness measurements. Individual data points presented; n = at least 10 per time point. The ventral tip is defined as the location where the posterior lobe is at its maximum height. The base was determined by moving 19.76 μm dorsally from the ventral tip. Statistical significance for each time point is indicated (unpaired t-test; ***p≤0.001; n.s. = not significant p≥0.05). (C–F) Comparison of *mCherry* RNAi (control) (C and E) and *dumpy*

*Figure 7 continued on next page*

*Figure 7 continued*

RNAi (D and F) at 44 hr APF and 52 hr APF labeled with with Dumpy:YFP (Green) and E-cadherin(Ecad; Magenta). GFP antibody was used to increase YFP signal. All cross-sections are oriented with apical side at the top and basal side at the bottom. Relevant structures labeled: Lateral plate (LP) posterior lobe (PL), and clasper (C). Cross-sections are max projections of 5.434 µm sections to show full Dumpy connection. Images were independently brightened to show relevant structures. Scale bar, 20 µm. n = at least five per experiment.

The online version of this article includes the following source data and figure supplement(s) for figure 7:

**Source data 1.** Individual measurements of *dumpy*-RNAi effects on the posterior lobe during its development.
**Figure supplement 1.** Variability in height of adult posterior lobe in *dumpy* knockdown.
**Figure supplement 2.** Remaining strands of Dumpy in *dumpy* knockdown.
**Figure supplement 3.** Correlation between VVA signal and *dumpy* knockdown phenotype.

aECM could alter cell signaling dynamics, as has been exhibited by the basal ECM (*Kirkpatrick et al., 2004*; *Kreuger et al., 2004*; *Wang et al., 2008*). Previous research has shown that the JAK/STAT pathway is important for posterior lobe development (*Glassford et al., 2015*), and the ability for a signal to reach its target cells could be altered in the absence of Dumpy. Of course, these models are not mutually exclusive and some combination of these mechanisms may be integrated to shape the posterior lobe. Our observations of increased cytoskeletal components in posterior lobe cells and the reduced height of cells that lack Dumpy in our knockdown experiments are consistent with all three mechanisms, which are difficult to differentiate experimentally.

## The role of aECM in the diversification of genital structures

Genitalia represents some of the most rapidly diversifying structures in the animal kingdom, and our results suggest the aECM may participate in the modification of *Drosophila* genital structures. The shape of the posterior lobe is extremely diverse among species of the *melanogaster* clade (*Coyne, 1989*). We confirmed that the extended posterior lobe of *D. sechellia* is associated with a similar aECM to *D. melanogaster* (*Figure 5—figure supplement 3*), confirming that this mechanism is likely common to other lobed species. Our results demonstrate that reducing the levels of Dumpy can affect the shape of the posterior lobe, with extreme knockdown phenotypes approximating the posterior lobe of *D. mauritiana*. Furthermore, the sheath and phallus show dense deposits of Dumpy, suggesting that the aECM could play important roles in diversifying these remarkably variable structures. During the course of evolution, one could imagine that by altering which cells are connected to the aECM, the physical nature of those connections, and the mechanical properties of those cells and/or their associated ECM could lead to changes in morphological shape. Hence, identifying genes that have evolved to differentiate these structures could uncover novel mechanisms for genetically controlling the behavior of this aECM and behaviors of cells bound to this dynamic scaffold.

## Integrating cells into a pre-existing aECM network to generate morphological novelty

Evolution is thought to act through the path of least resistance. When confronted with the remarkable diversity of genital morphologies present in insects, one must wonder how the intricate projections, bumps, and divots form through the action of epithelial rudiments. Traditionally the field of evo-devo research has focused on cell intrinsic processes that contribute to tissue morphogenesis, but here we show that the extrinsic forces from the aECM may be important for the evolution of a novelty. The aECM, while understudied, has been implicated in the morphogenesis of many structures (*Bailles et al., 2019*; *Diaz-de-la-Loza et al., 2018*; *Dong et al., 2014*; *Etournay et al., 2015*; *Fernandes et al., 2010*; *Heiman and Shaham, 2009*; *Low et al., 2019*; *Münster et al., 2019*; *Ray et al., 2015*; *Rosa et al., 2018*), making it a promising target for evolution. We find a conserved aECM network associated with central genital structures (clasper, sheath, and phallus) in both lobed and non-lobed species. However, on the dorsal side of the lateral plate we observed differences, with lobed species having substantial deposits of aECM that fill this area between the lateral plate and clasper and non-lobed species forming only thin strand-like connections (*Figure 8*). Because a complex network of aECM already existed on the genitalia to potentially pattern other structures, such as the phallus and its multiple elaborations (*Kamimura, 2010*; *Peluffo et al., 2015*; *Rice et al., 2019*), we hypothesize that this spatial expansion of aECM to the posterior lobe cells allowed them

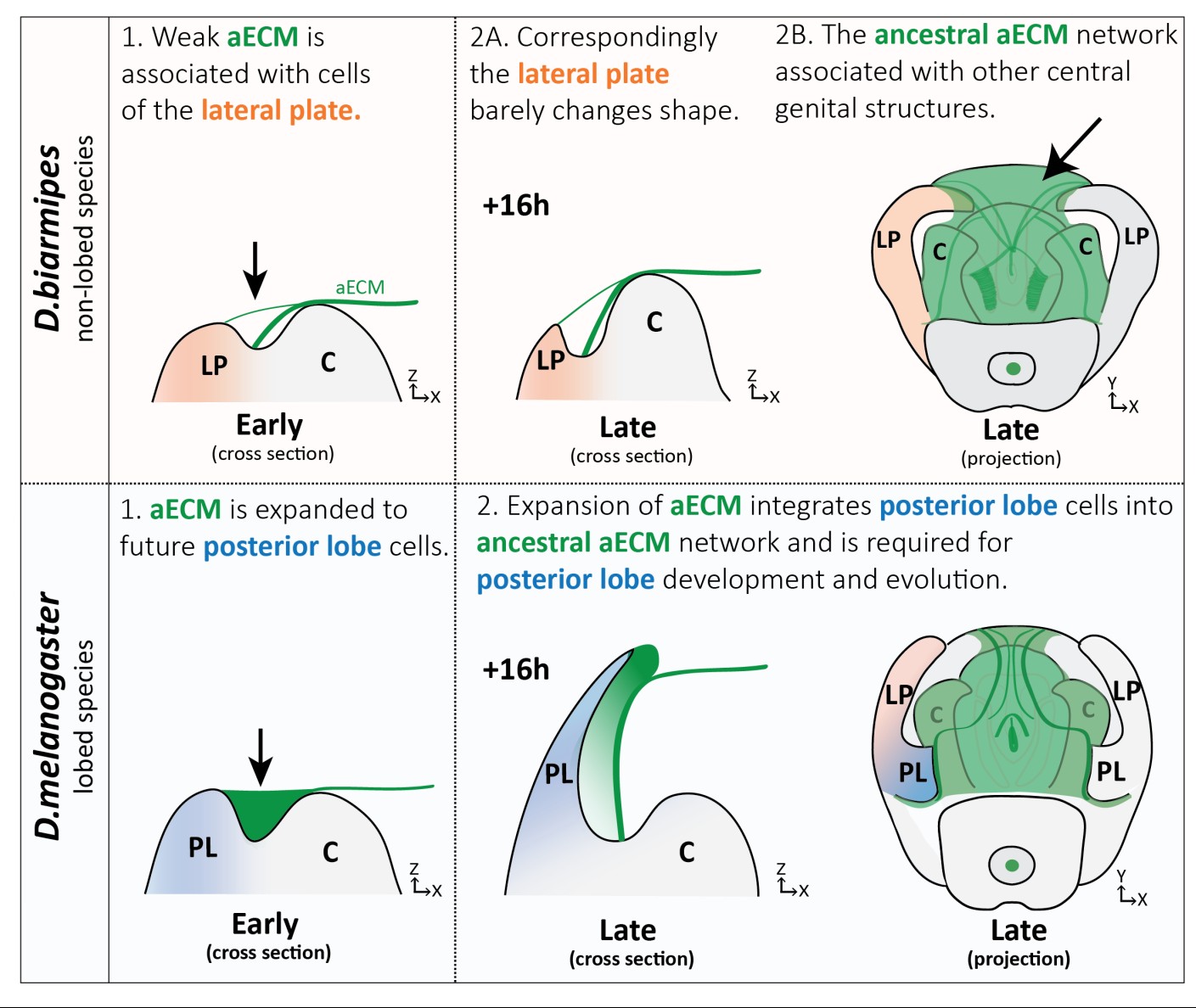

**Figure 8.** Expansion of aECM associated with the evolution of a morphological novelty. (Top) Illustration of non-lobed species, *D. biarmipes*, with ancestral aECM network covering central genital structures (2B) including the clasper (C), sheath, and phallus. Weak connections of aECM span from the clasper to the lateral plate (LP) during early development (1 and 2A - top). (Bottom) Illustration of lobed species, *D. melanogaster*. The aECM network has expanded to fill the crevice between the lateral plate and clasper (1-bottom) integrating these cells into the ancestral aECM network (2-bottom). This aECM population is needed for cells to properly project from the lateral plate, forming the posterior lobe.

The online version of this article includes the following figure supplement(s) for figure 8:

**Figure supplement 1.** Dumpy anchors posterior spiracles to surrounding cuticle.

to be integrated into the ancestral aECM network (*Figure 8*), a step which was likely significant to its evolution. Overall, this suggests that the aECM could be an unexpected target for generating novel anatomical structures. On the other hand, tissues which lack such an ancestral aECM network may be less likely to evolve projections through this mechanism. In addition, while Dumpy may be required for the development of the posterior lobe, additional components of the aECM, including factors that remodel the aECM or receptors that anchor the aECM to the cells, are likely also needed. Furthermore, we envision that additional processes may also be contributing to the full morphogenesis of the posterior lobe, including potential intrinsic processes that may contribute to

the cell shape changes we observe, such as the elevated concentrations of cytoskeletal components (*Figure 3*).

The expanded *dumpy* expression we discovered caused us to consider how the posterior lobe gained this aECM attachment. Models of gene network co-option have been appealing because they establish pre-existing mechanisms in place that can be rapidly ported to new locations to generate massive changes in a tissue. Interestingly, our previous work uncovered a gene regulatory network (GRN) that regulates development of an ancestral embryonic structure, the posterior spiracles, which was co-opted during the evolution of the posterior lobe and regulates its development (*Glassford et al., 2015*). Along these lines, *dumpy* is expressed in the developing posterior spiracles (*Wilkin et al., 2000*), and we have observed a thin strand of Dumpy:YFP connecting the posterior spiracles to the surrounding embryonic cuticle, known as the vitelline membrane (*Figure 8—figure supplement 1*; *Margaritis et al., 1980*). Identification of regulatory elements which activate *dumpy* in the posterior lobe will be necessary to determine whether its role in the posterior spiracle was relevant to the evolution of expanded genital expression.

Identifying the genetic changes that regulate cellular processes required for the evolution of new structures represents a major looming challenge in evo-devo research. Taken as a whole, the posterior lobe offers a promising model to begin to understand how gene regulatory networks and terminal effectors are connected, in addition to understanding how these networks become associated with novelties during evolution. As our understanding of the patterning of these tissues both at the level of transcriptional regulators (*Vincent et al., 2019*) and terminal effectors (such as Dumpy) becomes more complete, we can begin to connect the regulation of gene expression with the developmental behaviors of different cell types. The mapping of regulatory enhancers through reporter assays will allow us to trace the evolutionary history of these connections (*Glassford et al., 2015*). Furthermore, in systems where sophisticated genetic modifications are possible, CRISPR/Cas9 can be used to test the phenotypic consequences of genetic variation in these regulatory enhancers by exchanging regions between species (*Xu et al., 2017a*). Coupling such manipulations to quantitative measures of morphogenesis will illuminate the root causes of differences in tissue architecture (*Smith et al., 2018*).

# Materials and methods

## Key resources table

| Reagent type (species) or resource | Designation | Source or reference | Identifiers | Additional information |
|---|---|---|---|---|
| Antibody | Monoclonal rat anti-alpha tubulin (tyrosinated) | MilliporeSigma | Millipore Cat# MAB1864, RRID:AB_2210391 | IHC (1:500) |
| Antibody | Monoclonal mouse anti-alpha tubulin (acetylated) | Sigma-Aldrich | Sigma-Aldrich Cat# T6793, RRID:AB_477585 | IHC (1:500) |
| Antibody | Monoclonal rat anti-Ecadherin | DSHB | DSHB Cat# DCAD2, RRID:AB_528120 | IHC (1:500) |
| Antibody | Monoclonal mouse anti-Fasciclin III | DSHB | DSHB Cat# 7G10 anti-Fasciclin III, RRID:AB_528238 | IHC (1:500) |
| Antibody | Polyclonal rabbit anti-histone H3 (phospho S10) | Abcam | Abcam Cat# ab5176, RRID:AB_304763 | IHC (1:50) |
| Antibody | Polyclonal goat anti-GFP | Abcam | Abcam Cat# ab6662, RRID:AB_305635 | IHC (1:300) |
| Lectin | fluorescein Vicia Villosa Lectin (VVA) | Vector Laboratories | Vector Laboratories Cat# FL-1231, RRID:AB_2336856 | IHC (1:200) |

*Continued on next page*

*Continued*

| Reagent type (species) or resource | Designation | Source or reference | Identifiers | Additional information |
|---|---|---|---|---|
| Chemical compound, drug | rhodamine phalloidin | Thermo Fisher Scientific | Thermo Fisher Scientific Cat# R415, RRID:AB_2572408 | IHC (1:200) |
| Strain, strain background (*Drosophila melanogaster*) | y$^1$w$^1$ *Drosophila melanogaster* | Bloomington *Drosophila* Stock Center | BDSC Cat# 1495, RRID:BDSC_1495 | |
| Strain, strain background (*Drosophila biarmipes*) | wild type | National *Drosophila* Species Stock Center (NDSSC) | NDSSC Stock #: 14023–0361.10 RRID:FlyBase_FBst0203870 | |
| Strain, strain background (*Drosophila ananassae*) | wild type | National *Drosophila* Species Stock Center (NDSSC) | NDSSC Stock #: 14024–0371.13 RRID:FlyBase_FBst0201380 | No longer available |
| Strain, strain background (*Drosophila pseudoobscura*) | wild type | National *Drosophila* Species Stock Center (NDSSC) | NDSSC Stock #: 14011–0121.87 RRID:FlyBase_FBst0200074 | No longer available |
| Strain, strain background (*Drosophila sechellia*) | Wild type | National *Drosophila* Species Stock Center (NDSSC) | NDSSC Stock #: #14021–0248.03 RRID:FlyBase_FBst0201190 | No longer available |
| Genetic reagent (*Drosophila melanogaster*) | *UAS*-Raeppli-CAAX | Bloomington *Drosophila* Stock Center (BDSC) | BDSC Cat# 55084, RRID:BDSC_55084 | |
| Genetic reagent (*Drosophila melanogaster*) | *Pox neuro-Gal4* | (**Boll and Noll, 2002**) | | Construct #13 |
| Genetic reagent (*Drosophila melanogaster*) | *D. simulans Pox neuro*-Gal4 | This paper | | Can be obtained from Mark Rebeiz, rebeiz@pitt.edu |
| Genetic reagent (*Drosophila melanogaster*) | hs – flippase[122] | Gift from Erika A. Bach | Flybase: FBtp0001101 | |
| Genetic reagent (*Drosophila melanogaster*) | *armadillo-GFP* | (**Huang et al., 2012**) | | |
| Genetic reagent (*Drosophila melanogaster*) | Dumpy:YFP | *Drosophila* Genomics and Genetic Resources | DGGR Cat# 115238, RRID:DGGR_115238 | |
| Genetic reagent (*Drosophila melanogaster*) | Viking:GFP | *Drosophila* Genomics and Genetic Resources | DGGR Cat# 110626, RRID:DGGR_110626 | |
| Genetic reagent (*Drosophila melanogaster*) | Perlecan:GFP | *Drosophila* Genomics and Genetic Resources | DGGR Cat# 110807, RRID:DGGR_ 110807 | |
| Genetic reagent (*Drosophila melanogaster*) | E-cadherin:mCherry | Bloomington *Drosophila* stock center | BDSC Cat# 59014, RRID:BDSC_59014 | |
| Genetic reagent (*Drosophila melanogaster*) | *UAS*-dumpyRNAi | Vienna *Drosophila* Resource Center | VDRC Cat#44029, RRID:FlyBase_FBst0465370 | |
| Genetic reagent (*Drosophila melanogaster*) | *UAS*-mCherryRNAi | Bloomington *Drosophila* stock center | BDSC Cat# 35785, RRID:BDSC_35785 | |

*Continued on next page*

*Continued*

| Reagent type (species) or resource | Designation | Source or reference | Identifiers | Additional information |
|---|---|---|---|---|
| Recombinant DNA reagent | pS3aG4 | Gift from Benjamin Prud'homme | | Gal4 vector used to make *D. simulans Pox neuro* gal4 line |
| Sequence-based reagent | GCCACTAACA ATCCATGCGGTT | This paper | | *dumpy* probe forward primer. Obtained from Integrated DNA Technologies. |
| Sequence-based reagent | TAATACGACTCACTATAG GGAGAAATAGCCCT GTCCTTGGAATCC | This paper | | *dumpy* probe reverse primer with T7 primer. Obtained from Integrated DNA Technologies. |
| Sequence-based reagent | TTCCGGGCGCGCCTCGG TGGCTTAACACGCGCATT | This paper | | *D. simulans Pox neuro* forward primer for gal four line. Obtained from Integrated DNA Technologies. |
| Sequence-based reagent | TTGCCCCTGCAGGATC GCTGATTCCATGGCCCAGT | This paper | | *D. simulans Pox neuro* reverse primer for gal four line. Obtained from Integrated DNA Technologies. |
| Software algorithm | Fiji (ImageJ v2.0) | (*Schindelin et al., 2012*) | RRID:SCR_002285 | |
| Software algorithm | GenePalette | (*Rebeiz and Posakony, 2004*; *Smith et al., 2017*) | | |
| Software algorithm | Leica Application Suite X | Leica | RRID:SCR_013673 | |
| Software algorithm | Microsoft Excel | Microsoft | RRID:SCR_016137 | |
| Software algorithm | MorphoGraphX | (*Barbier de Reuille et al., 2015*) | | |
| Software algorithm | Prism 8 | GraphPad | RRID:SCR_002798 | |

## Fly stocks and genetics

Fly stocks were reared using standard culture conditions. Wild type species used in this study were obtained from the University of California, San Diego *Drosophila* Stock Center (now known as The National *Drosophila* Species Stock Center at Cornell University) (*Drosophila biarmipes* #14024–0361.10, *Drosophila ananassae* #14024–0371.13, *Drosophila pseudoobscura* #14011–0121.87) and from the Bloomington *Drosophila* Stock Center (*Drosophila melanogaster* [$y^1w^1$] #1495). *Pox neuro*-Gal4 (construct #13) was obtained from Werner Boll (*Boll and Noll, 2002*). The following were obtained from the Bloomington*Drosophila*stock center: *UAS*-Raeppli-CAAX (#55084), *armadillo-GFP* (#8556), Ecadherin:mCherry (#59014), and *UAS*-mCherryRNAi (control for RNAi experiments, as mCherry is not present in the *Drosophila* genome) (#35785). *UAS-dumpy*-RNAi was obtained from the Vienna *Drosophila* Resource Center (#44029) and Dumpy:YFP was obtained from the *Drosophila* Genomics and Genetic Resources (#115238).

For the Raeppli experiments, stable lines of hs-flippase;;*UAS*-Raeppli-CAAX/*UAS*-Raeppli-CAAX and *D. simulans Pox neuro*-gal4/*D. simulans Pox neuro*-Gal4;*UAS*-Raeppli-CAAX/*UAS*-Raeppli-CAAX were generated. *D. simulans Pox neuro*-Gal4 was used as opposed to *Pox neuro*-Gal4 because a Gal4 driver on the second chromosome was required. Virgin females from the first line were crossed to males from the second line to ensure hs-flippase was inherited by all offspring. Offspring were collected and grown as normal, heat shocked at 37°C for 1 hr around 24 to 28 hr APF, and allowed to finish development at 25°C. Although this system is designed to label cells with multiple colors (*Kanca et al., 2014*), we could only detect mTFP1 in heat-shocked tissues which we suspect is caused by insufficient activation of the other fluorescent reporters to detectible levels.

## Sample preparation

Pupal samples were prepared following the protocol in *Glassford et al. (2015)*. Briefly, samples were incubated at 25°C unless otherwise noted (*Glassford et al., 2015*). Dissections were performed in cold PBS, pupae were cut in half, removed from their pupal cases, and fat bodies removed by flushing, leaving the genitalia connected to the surrounding pupal cuticle. Larval samples were dissected in cold PBS by cutting the larva in half, and flipping the posterior end of the larva inside out. All samples were fixed for 30 min at room temperature in PBS with 0.1% Triton X-100% and 4% paraformaldehyde. Samples stained with phalloidin had Triton X-100 concentrations increased to 0.3%. Samples used for VVA staining were removed from pupal cuticle before being fixed in PBS with 0.1% Triton X-100, 4% paraformaldehyde, and 1% trichloroacetic acid on ice for 1 hr followed by 30 min at room temperature. The trichloroacetic acid method causes some slight tissue distortion, as the precipitation treatment utilized to refine the VVA signal causes the posterior lobe to become slightly deformed and curve in towards the clasper. However, similar defects were not observed in the other structures such as the lateral plate or in *D. biarmipes*. Samples were stored in PBT for immunostaining at 4°C for up to two days. For in situ hybridization, samples were rinsed twice in methanol and rinsed twice in ethanol. Samples were stored at −20°C in ethanol.

## Immunostaining and in situ hybridization

For immunostaining, genital samples were removed from the surrounding pupal cuticle and incubated overnight at 4°C with primary antibodies diluted in PBS with 0.1% Triton-X (PBT). VVA and phalloidin samples were placed on a rocker. The following primary antibodies were used: rat anti-alpha tubulin (tyrosinated) 1:500 (MAB 1864-I, MilliporeSigma), mouse anti-alpha tubulin (acetylated) 1:500 (T6793, Sigma-Aldrich), rat anti-Ecadherin 1:500 (DCAD2, DSHB), mouse anti-Fasciclin III 1:500 (7G10, DSHB), rabbit anti-histone H3 (phospho S10) 1:50 (ab5176, Abcam), goat anti-GFP 1:300 (ab6662, Abcam), fluorescein Vicia Villosa Lectin (VVA) 1:200 (FL-1231, Vector Laboratories). The goat anti-GFP was used to increase signal of Dumpy:YFP in the knockdown experiments only. Primary antibody was removed by performing two quick rinses and two long washes (at least 5 min) in PBT. Samples were incubated overnight at 4°C in secondary antibodies diluted in PBT. The following secondary antibodies were used: donkey anti-rat Alexa 594 1:500 (A21209, Invitrogen), donkey anti-mouse Alexa 488 1:500 (A21202, Thermo Fisher Scientific), donkey anti-rat Alexa 488 1:500 (A21208, Thermo Fisher Scientific), goat anti-mouse Alexa 594 1:500 (A-11005, Thermo Fisher Scientific), goat anti-rabbit Alexa 594 1:500 (A-11012, Thermo Fisher Scientific), donkey anti-goat Cy2 1:500 (705-225-147, Jackson ImmunoResearch). Rhodamine phalloidin (R415, Thermo Fisher Scientific) stain was performed with secondary antibody. Samples were washed out of secondary antibody by performing two quick rinses and two long washes (at least 5 min) in PBT. Samples were then incubated in 50% PBT/50% glycerol solution for at least 5 min. Pupal samples were mounted on glass slides coated with Poly-L-Lysine Solution. Glass slides had 1 to 2 layers of double sided tape with a well cut out in which the sample was placed and covered with a cover slip. in situ hybridization was performed following the protocol in *Rebeiz et al. (2009)* with modifications to perform in situs in the InsituPro VSi robot (Intavis Bioanalytical Instruments) as done by Glassford et al., with the exception that the genital samples were removed from the surrounding pupal cuticle before placing in the InsituPro VSi robot (*Glassford et al., 2015*; *Rebeiz et al., 2009*).

## Microscopy and live imaging

Cuticles of adult posterior lobes and in situ hybridization samples were imaged on a Leica DM2000 microscope with a 40x objective for cuticles and a 10x objective for in situ samples. Samples with fluorescent antibodies and fluorescently tagged proteins were imaged using a Leica TCS SP5 Confocal microscope with either a 40x or 63x oil immersion objective.

To live-image genital development, a 2% agar solution was poured into a small petri dish, filling the dish half way. A 0.1–10 μL pipette tip was used to make small wells in the agar for pupal samples. Timed pupal samples were inserted head first into the small well and a 20–200 μL pipette tip was used to push samples into agar by placing the tip around the posterior spiracles on the pupal case. To better image the developing genitalia, the pupal case at the posterior end was removed with forceps. Deionized water was used to cover the samples and imaged on a Leica TCS SP5 Confocal microscope using a 63x water objective.

To live-image embryos, Dumpy:YFP flies were grown in egg-laying chamber with grape agar plates (Genesee Scientific). Embryos were removed from plates using forceps and rolled on a piece of double sided tape to remove the chorion. Embryos then were positioned on a glass coverslip coated with embryo glue. A glass slide was covered with double sided tape and a well was made and filled with halocarbon 27 oil. The cover slip with the embryos was then placed on the glass slide, submerging the embryos in halocarbon oil. Embryos were imaged on a Leica TCS SP8 confocal with a 63x oil objective.

## Image analysis

Images were processed with Fiji (*Schindelin et al., 2012*) and Photoshop. Three-dimensional views were rotated and captured in MorphoGraphX (*Barbier de Reuille et al., 2015*) or Leica Application Suite X. Movies were processed in Fiji and cell rearrangements were tracked using the manual tracking plugin. Tissue thickness/cell height during development was measured in cross-sections by drawing a line centered between the two sides (based on apical membrane) of the lobe until the basal side was reached. Area of adult posterior lobe cuticles and height of the adult lobe were measured by using the lateral plate as a guide for determining the bottom boundary of the posterior lobe. To prevent any possible bias for one lobe vs the other (i.e. left vs right) which lobe was used in statistical analysis was randomly decided, except for *Figure 6—figure supplement 1* where both posterior lobes were considered.

## Transgenic constructs

To make the *D. simulans Pox neuro*-Gal4 driver, the posterior lobe enhancer of *Pox neuro* in *D. simulans,* identified in *Glassford et al. (2015)*, was cloned using primers listed in the key resources table from genomic DNA purified with the DNeasy Blood and Tissue Kit (QIAGEN). Primers were designed using sequence conservation with the GenePalette software tool (*Rebeiz and Posakony, 2004*; *Smith et al., 2017*). The cloned sequence was inserted into the pS3aG4 (Gal4) using *AscI* and *SbfI* restriction sites. The final construct was inserted into the 51D landing site on the second chromosome (*Bischof et al., 2007*).

# Acknowledgements

The authors thank the members of the M.R. laboratory for comments and discussion on the manuscript. We thank Werner Boll and Markus Noll for the *Pox neuro*-gal4 line, the Bloomington stock center, VDRC, and DGGR stock centers for fly stocks, Benjamin Prud'homme for the s3aG4 vector, Erika A, Bach for the hs-flippase line, and Winslow Johnson for the *D. simulans Pox neuro*-gal4 line. This work was supported by the National Institutes of Health (GM107387 to M.R. and HD044750 to LAD).

# Additional information

## Funding

| Funder | Grant reference number | Author |
| --- | --- | --- |
| National Institutes of Health | GM107387 | Mark Rebeiz |
| National Institutes of Health | HD044750 | Lance A Davidson |

The funders had no role in study design, data collection and interpretation, or the decision to submit the work for publication.

## Author contributions

Sarah Jacquelyn Smith, Conceptualization, Data curation, Formal analysis, Validation, Investigation, Visualization, Methodology, Writing - original draft, Writing - review and editing; Lance A Davidson, Conceptualization, Supervision, Methodology, Writing - review and editing; Mark Rebeiz, Conceptualization, Supervision, Funding acquisition, Methodology, Writing - review and editing

Author ORCIDs
Sarah Jacquelyn Smith (iD) https://orcid.org/0000-0002-1469-1821
Lance A Davidson (iD) http://orcid.org/0000-0002-2956-0437
Mark Rebeiz (iD) https://orcid.org/0000-0001-5731-5570

Decision letter and Author response
Decision letter https://doi.org/10.7554/eLife.55965.sa1
Author response https://doi.org/10.7554/eLife.55965.sa2

## Additional files

### Supplementary files

• Transparent reporting form

### Data availability

All data generated or analyzed during this study are included in the manuscript and supporting files. Source data files have been provided for Figures 2, 6, 7.

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
