## [Decision Letter]

**Acceptance summary:**

The manuscript by Smith and colleagues examines the developmental basis of a novel structure and how it may have evolved. The authors focus on the posterior lobe of *Drosophila melanogaster*, a structure that recently evolved in this species and that is absent from closely related Drosophilids. Through a series of experiments, the authors show that the posterior lobe is formed largely through increases in the height of cells, rather than through cell proliferation or cell intercalation. Furthermore, the authors are able to attribute this change in cell size to the presence of the apical extracellular matrix (aECM) and in particular to the aECM protein Dumpy. These results suggest an important role for aECM in the evolution of genital structures in Drosophilids.

**Decision letter after peer review:**

[Editors’ note: the authors submitted for reconsideration following the decision after peer review. What follows is the decision letter after the first round of review.]

Thank you for submitting your work entitled "Expansion of apical extracellular matrix underlies the morphogenesis of a recently evolved structure" for consideration by *eLife*. Your article has been reviewed by 3 peer reviewers, and the evaluation has been overseen by a Reviewing Editor and a Senior Editor. The reviewers have opted to remain anonymous.

Our decision has been reached after consultation between the reviewers. Based on these discussions and the individual reviews below, we regret to inform you that your work will not be considered further for publication in *eLife* in its present form. As you will see from the reviews, there was consensus that the topic addressed is of importance and that the work presented is an important first step toward elucidating the molecular basis of posterior lobe diversity. The key concern is that the data presented don't yet "prove" that the Dumpy aECM causes the posterior lobe diversity. For example, either reviewer #2's fourth point (how does the VVA-labeled aECM differ in other lobed species versus *melanogaster*?) or reviewer's #3 suggestion (doing the complementary gain-of-function experiment) would allow you to explore in more depth the connection between the aECM and posterior-lobe diversity. Recognizing that these experiments entail a lot more experimental*eLife*, we have opted to reject the manuscript. But we would be open to considered a revised manuscript that addresses this key concern as well as the reviewers' other concerns.

Reviewer #1:

The manuscript by Smith and colleagues examines the developmental basis of a novel structure and how it may have evolved. The authors focus on the posterior lobe of *Drosophila melanogaster*, a structure that recently evolved in this species and that is absent from closely related Drosophilids. Through a series of experiments, the authors show that the posterior lobe is formed largely through increases in the height of cells, rather than through cell proliferation or cell intercalation. Furthermore, the authors are able to attribute this change in cell size to the presence of the apical extracellular matrix (aECM) and in particular to the aECM protein Dumpy. These results suggest an important role for aECM in the evolution of genital structures in Drosophilids.

The manuscript seems well written, the experiments appear to be well done, and the topic and question are of importance to evolutionary developmental biology.

Reviewer #2:

In this manuscript, Smith et al., investigate the morphological basis for a recently evolved and novel structure, the male posterior lobe in *Drosophila melanogaster*. The posterior lobe (PL), an outgrowth of the genitalia required for copulation, only appears in the *melanogaster* clade of *Drosophila*, and not in more distantly related species. The authors image the PL from the lateral plate at specific stages in *D. melanogaster* versus a non-lobed species, *D. biarmipes*. The growth of the PL during development and splitting from the lateral plate does not come from cell proliferation or cell intercalation. Instead, the authors propose that PL growth derives from cell lengthening; a membrane marker shows that cells span the length of the basal to apical surface of the PL. Thickness of the PL changes during development, suggesting that PL tissue development is dynamic. Next, the authors investigate the cellular and extracellular basis for PL morphogenesis. The PL is enriched for F-actin and microtubules, suggesting that the cytoskeleton may contribute to the shape and growth. However, the authors discover that a Dumpy-YFP-enriched apical extracellular matrix (aECM) is associated with the developing PL, particularly at the apex of the PL at late stages. Further, Dumpy RNA expression is expanded to the developing PL in lobed but not non-lobed species. The authors then show that VVA lectin resembles Dumpy-YFP aECM. This aECM network differs in non-lobed species compared to *D. melanogaster*, suggesting that the Dumpy-aEM helps shape PL morphogenesis. Lastly, the authors demonstrate that the PL is reduced and changes shape when Dumpy is knocked down by RNAi in *D. melanogaster*. PL changes correlate with altered Dumpy-YFP aECM organization. The authors conclude that the aECM is required for PL development in lobed species, but may have different functions in genital morphogenesis of non-lobed species.

Overall, this is a well-written manuscript, with very clear imaging data and videos. This is an important topic of general interest to developmental biologists, especially those interested in the evolutionary basis of tissue morphology. To strengthen the conclusions, however, clarification of several points would be helpful.

First, the authors use the Raeppli system to label cells with a membrane marker and show that the PL is one-cell thick, which continues to grow/lengthen. The authors could clarify why they used this technique specifically, rather than a regular FLP-out clone of a membrane-tethered GFP (e.g. UAS-PLCδPH-GFP or UAS-mCD8-GFP). I'm also concerned that the N's are low (n=4 clones). Did the authors label and image additional cells of the PL, and at different stages of development, to more convincingly show that the epithelium is only one cell layer thick throughout the breadth of the tissue?

Second, the authors use VVA-FITC to label the aECM in other *Drosophila* species. Can the authors confirm that VVA labels the same aECM structures as Dumpy-YFP in *D. melanogaster*? If so, this would go a long way to validate VVA as a great proxy for Dumpy. It looks like there is one commercial source of VVL/VVA-TRITC (Glycomatrix.com) that could be used.

Third, the authors look at Dumpy-YFP in a Dumpy RNAi background (Figure 7C-F). Shouldn't the RNAi target Dumpy-YFP and thus reduce its expression? This could change the interpretation of how the aECM associates with the ventral portion of the PL. Visualizing VVA here could strengthen this association.

Fourth, the PL appears quite different within the *melanogaster* clade (e.g. Glassford et al., 2015 shows the simulans has a very large PL and mauritiana a very small PL). How does the VVA-labeled aECM differ in these (or another) lobed species versus *melanogaster*? The similarities/differences could really strengthen the association of aECM organization and how the PL shape develops in lobed versus non-lobed species.

Fifth, the authors conclude that Dumpy is required for proper posterior lobe formation. The data seems to support a role for Dumpy in sculpting and shaping the lobe, as the PL still forms in the dumpy RNAi animals. Or do the authors think that there is residual Dumpy in the RNAi condition that may account for the PL still forming, albeit as a smaller tissue with altered shape? Alternatively, does the GAL4 driver used to knockdown Dumpy have variable expression that can account for the variability in the PL phenotypes?

Lastly, the authors could further discuss the possibility that intrinsic factors such as the cytoskeleton may contribute to PL morphogenesis. The phenotypes caused by Dumpy RNAi are not completely penetrant (e.g. the PL still forms, but has a cleft). Do the authors think that a combination of extrinsic aECM along with cytoskeletal changes contribute? Is there any role for apical cell constriction of the epithelia, for example, in forming the crevice between the PL and clasper, and that this could help the PL form in these species?

Reviewer #3:

Smith et al., present a thorough analysis of the morphogenesis of the posterior lobe of the *Drosophila melanogaster* male genitalia, and compare this morphogenetic process with non-lobed yet relatively closely related species. They find that the posterior lobe forms by a major increase in cell height; that the developing genitalia associate with an intricate apical extracellular matrix; and that a component of this matrix, Dumpy, has expanded in expression in lobed species and is required for lobe development in *D. melanogaster*. This work joins other recent work (reviewed in the Introduction) that is beginning to reveal the cellular-developmental basis of evolutionary change, and as the authors note their work is an important addition because of its focus on a novel structure rather than diversification of an ancestral structure.

The experiments appear to have been well conceived and rigorously executed, and the information they provide is an important starting point for understanding the evolution of this model novelty (both its origin and its subsequent diversification).

Although this is solid work, I do have reservations about its ultimate impact. It is not clear whether changes in Dumpy expression were at all causative in the evolution of the posterior lobe. It is not entirely surprising that disrupting Dumpy, which attaches to the developing lobe, disrupts its ultimate morphology. What is missing is the complementary gain-of-function experiment: does expanding Dumpy expression in a non-lobed species create anything akin to a new lobe? Granted, this kind of experiment might not be feasible in a non-model fly, and if it could be done it might yield a hard-to-interpret negative result. But then the question remains: how do we solve the "major looming challenge in evo-devo research" that the authors note in their final sentence Discussion section) – what are the genetic changes underlying the cellular changes? Moving in that direction would indeed be a major advance, and it seems to me that the ultimate impact of the present work lies in how possible it will be to uncover the relevant genetic changes underlying the novelty. I don't have any specific experiments to suggest (and I'd be reluctant to suggest major new experiments anyway), but the uncertainty about making progress does constrain my enthusiasm somewhat. And in the meantime, the title ("Expansion of aECM underlies the morphogenesis.…") is a bit misleading and should reflect more this uncertainty.

---

## [Author Response]

[Editors’ note: the authors resubmitted a revised version of the paper for consideration. What follows is the authors’ response to the first round of review.]

The paper has been greatly improved in response to the reviewer feedback. We have performed additional experiments and added significant data to address nearly

every reviewer comment and concern. Although one requested experiment was not

addressable, we detail in our response how this suggested experiment is infeasible, and

describe the current practices and theoretical frameworks of the evo-devo field upon which our conclusions are soundly based.

Reviewer #1:The manuscript by Smith and colleagues examines the developmental basis of a novel structure and how it may have evolved. The authors focus on the posterior lobe of *Drosophila melanogaster*, a structure that recently evolved in this species and that is absent from closely related Drosophilids. Through a series of experiments, the authors show that the posterior lobe is formed largely through increases in the height of cells, rather than through cell proliferation or cell intercalation. Furthermore, the authors are able to attribute this change in cell size to the presence of the apical extracellular matrix (aECM) and in particular to the aECM protein Dumpy. These results suggest an important role for aECM in the evolution of genital structures in Drosophilids.The manuscript seems well written, the experiments appear to be well done, and the topic and question are of importance to evolutionary developmental biology.

We thank the reviewer for their comments and perspective on our work.

Reviewer #2:In this manuscript, Smith et al., investigate the morphological basis for a recently evolved and novel structure, the male posterior lobe in *Drosophila melanogaster*. The posterior lobe (PL), an outgrowth of the genitalia required for copulation, only appears in the melanogaster clade of Drosophila, and not in more distantly related species. The authors image the PL from the lateral plate at specific stages in *D. melanogaster* versus a non-lobed species, *D. biarmipes*. The growth of the PL during development and splitting from the lateral plate does not come from cell proliferation or cell intercalation. Instead, the authors propose that PL growth derives from cell lengthening; a membrane marker shows that cells span the length of the basal to apical surface of the PL. Thickness of the PL changes during development, suggesting that PL tissue development is dynamic. Next, the authors investigate the cellular and extracellular basis for PL morphogenesis. The PL is enriched for F-actin and microtubules, suggesting that the cytoskeleton may contribute to the shape and growth. However, the authors discover that a Dumpy-YFP-enriched apical extracellular matrix (aECM) is associated with the developing PL, particularly at the apex of the PL at late stages. Further, Dumpy RNA expression is expanded to the developing PL in lobed but not non-lobed species. The authors then show that VVA lectin resembles Dumpy-YFP aECM. This aECM network differs in non-lobed species compared to D. melanogaster, suggesting that the Dumpy-aEM helps shape PL morphogenesis. Lastly, the authors demonstrate that the PL is reduced and changes shape when Dumpy is knocked down by RNAi in D. melanogaster. PL changes correlate with altered Dumpy-YFP aECM organization. The authors conclude that the aECM is required for PL development in lobed species, but may have different functions in genital morphogenesis of non-lobed species.Overall, this is a well-written manuscript, with very clear imaging data and videos. This is an important topic of general interest to developmental biologists, especially those interested in the evolutionary basis of tissue morphology. To strengthen the conclusions, however, clarification of several points would be helpful.

Thank you for the very careful reading of our manuscript and appreciation of the topic and our conclusions. We have addressed all of the comments raised below with additional experiments and modifications.

First, the authors use the Raeppli system to label cells with a membrane marker and show that the PL is one-cell thick, which continues to grow/lengthen. The authors could clarify why they used this technique specifically, rather than a regular FLP-out clone of a membrane-tethered GFP (e.g. UAS-PLCδPH-GFP or UAS-mCD8-GFP). I'm also concerned that the N's are low (n=4 clones). Did the authors label and image additional cells of the PL, and at different stages of development, to more convincingly show that the epithelium is only one cell layer thick throughout the breadth of the tissue?

We have increased our N for this experiment to 10. We had initially sought to use the Raeppli system to label multiple cells in different colors. While this did not pan out in our hands (likely due to insufficient activation of the reporters such that only one color was detected), it did allow us to measure the height of single cells. We also note that our staining of microtubules across a variety of timepoints shows apicobasal polarity consistent with observations from the Raeppli experiment. We explain the rationale for our approach in the Methods section:

“Although this system is designed to label cells with multiple colors (Kanca et al., 2014), we could only detect mTFP1 in heat-shocked tissues which we suspect is caused by insufficient activation of the other fluorescent reporters to detectible levels.” (subsection “Immunostaining and in situ hybridization”).

Second, the authors use VVA-FITC to label the aECM in other Drosophila species. Can the authors confirm that VVA labels the same aECM structures as Dumpy-YFP in *D. melanogaster*? If so, this would go a long way to validate VVA as a great proxy for Dumpy. It looks like there is one commercial source of VVL/VVA-TRITC (Glycomatrix.com) that could be used.

We tried a variety of lectins, including VVA-Texas Red and carefully compared these to DumpyYFP. Of these, VVA-FITC showed the most sharp/reproducible pattern that matched the Dumpy-YFP experiment. We now document more completely how this was determined in Figure 5, supplement 2, where we show how diagnostic aspects of the Dumpy pattern are mirrored by VVA-FITC. We also show in Figure 7—figure supplement 3 how the VVA signal is altered during Dumpy knockdown, further confirming that these signals are related by more than just a simple correlation. We acknowledge that the aECM is likely complex, and VVA almost certainly labels more than just Dumpy protein. However, our experiments suggest that VVA is a useful proxy for aECM, a previously unseen feature of genital development that can now be compared between species using this method.

Third, the authors look at Dumpy-YFP in a Dumpy RNAi background (Figure 7C-F). Shouldn't the RNAi target Dumpy-YFP and thus reduce its expression? This could change the interpretation of how the aECM associates with the ventral portion of the PL. Visualizing VVA here could strengthen this association.

We agree that Dumpy RNAi will target both wildtype and YFP tagged Dumpy in this experiment, and we realize that the previous wording had not made this point completely clear. Indeed, we sought to image Dumpy-YFP in the RNAi background to more carefully assess how the phenotype of Dumpy-RNAi manifests. That we see remaining connections in the knockdown confirms that the knockdown is not absolute, and shows how the tallest cells of the defective lobe are nevertheless those which maintain a connection with Dumpy-positive aECM. We modified the text as such to make the setup of this experiment more clear:

“Why would dorsal cells of the posterior lobe be more affected in a dumpy knockdown? We hypothesized that our posterior lobe specific driver is not strong enough to remove all deposits of Dumpy associated with the posterior lobe. To better understand this, we examined Dumpy:YFP localization in the dumpy knockdown background, in which both dumpy and dumpy:yfp should be targeted by the RNAi treatment.” (Subsection “Correlation of Dumpy deposition and cell height in the posterior lobe”.)

To address the reviewer’s point, we now show the VVA stain of Dumpy-RNAi in Figure 7 supplement 3. This experiment confirms that these remaining connections between the tallest portion of the posterior lobe and the aECM are the only obvious connections in the Dumpy-RNAi experiment.

Fourth, the PL appears quite different within the melanogaster clade (e.g. Glassford et al., 2015 shows the simulans has a very large PL and mauritiana a very small PL). How does the VVA-labeled aECM differ in these (or another) lobed species versus melanogaster? The similarities/differences could really strengthen the association of aECM organization and how the PL shape develops in lobed versus non-lobed species.

This is a great point. We see such an experiment as a useful demonstration that this mechanism is likely common to all lobed species, representing a morphogenetic process that coincides with the origin of this structure. As per the reviewer’s request, we have added an experimental stain of VVA to show that it is associated with the posterior lobe of another species (*D. sechellia*) whose morphology differs from *D. melanogaster* (Figure 5—figure supplement 3). Although the association correlates with the unique shape of the lobe in this species, we do feel that it would be an over-interpretation to conclude that this demonstrates that the aECM causes differences in lobe morphology. Studying such a problem would require a very different type of study of the genetic variants between these species, which is ongoing. However, the data do demonstrate that this is not a mechanism that only evolved in *D. melanogaster*.

Fifth, the authors conclude that Dumpy is required for proper posterior lobe formation. The data seems to support a role for Dumpy in sculpting and shaping the lobe, as the PL still forms in the dumpy RNAi animals. Or do the authors think that there is residual Dumpy in the RNAi condition that may account for the PL still forming, albeit as a smaller tissue with altered shape? Alternatively, does the GAL4 driver used to knockdown Dumpy have variable expression that can account for the variability in the PL phenotypes?

This is exactly what we think – when we knock down Dumpy, there are still Dumpy-YFP connections between the remaining tallest cells of the lobe and the centrally localized deposits of aECM (Figure 7D). These deposits of Dumpy in the genitalia are highly concentrated, and we have been unable to fully eliminate them by RNAi using a variety of drivers. We thus interpret the variability in *dumpy-RNAi* phenotype to be most consistent with variable efficiency of knockdown. We have revised the text to clarify this view:

“Why would dorsal cells of the posterior lobe be more affected in a dumpy knockdown? We hypothesized that our posterior lobe specific driver is not strong enough to remove all deposits of Dumpy associated with the posterior lobe.” (subsection “Correlation of Dumpy deposition and cell height in the posterior lobe”)

Lastly, the authors could further discuss the possibility that intrinsic factors such as the cytoskeleton may contribute to PL morphogenesis. The phenotypes caused by Dumpy RNAi are not completely penetrant (e.g. the PL still forms, but has a cleft). Do the authors think that a combination of extrinsic aECM along with cytoskeletal changes contribute? Is there any role for apical cell constriction of the epithelia, for example, in forming the crevice between the PL and clasper, and that this could help the PL form in these species?

We agree with the reviewer that multiple mechanisms are likely acting. We recognize the possible roles for intrinsic factors at multiple places: Results section, Discussion section:

“Furthermore, we envision that additional processes may also be contributing to the full morphogenesis of the posterior lobe, including potential intrinsic processes that may contribute to the cell shape changes we observe, such as the concentrations of cytoskeletal components we observed (Figure 3).”(subsection “Integrating cells into a pre-existing aECM network to generate morphological novelty”)

However, we focused our discussion primarily on the aECM, as very few studies have investigated extrinsic factors in the morphogenesis of epithelial structures, and comparative evolutionary studies of extrinsic factors are particularly rare. Thus, we felt that the focus on the aECM best emphasized the significance of our work, while fully acknowledging the existence of intrinsic mechanisms.

Reviewer #3:Smith et al., present a thorough analysis of the morphogenesis of the posterior lobe of the *Drosophila melanogaster* male genitalia, and compare this morphogenetic process with non-lobed yet relatively closely related species. They find that the posterior lobe forms by a major increase in cell height; that the developing genitalia associate with an intricate apical extracellular matrix; and that a component of this matrix, Dumpy, has expanded in expression in lobed species and is required for lobe development in *D. melanogaster*. This work joins other recent work (reviewed in the Introduction) that is beginning to reveal the cellular-developmental basis of evolutionary change, and as the authors note their work is an important addition because of its focus on a novel structure rather than diversification of an ancestral structure.The experiments appear to have been well conceived and rigorously executed, and the information they provide is an important starting point for understanding the evolution of this model novelty (both its origin and its subsequent diversification).

We thank the reviewer for their comment on the rigor and importance of our studies, as well as the thoughtful reactions and suggestions below.

Although this is solid work, I do have reservations about its ultimate impact. It is not clear whether changes in Dumpy expression were at all causative in the evolution of the posterior lobe. It is not entirely surprising that disrupting Dumpy, which attaches to the developing lobe, disrupts its ultimate morphology. What is missing is the complementary gain-of-function experiment: does expanding Dumpy expression in a non-lobed species create anything akin to a new lobe? Granted, this kind of experiment might not be feasible in a non-model fly, and if it could be done it might yield a hard-to-interpret negative result. But then the question remains: how do we solve the "major looming challenge in evo-devo research" that the authors note in their final sentence Discussion section) – what are the genetic changes underlying the cellular changes? Moving in that direction would indeed be a major advance, and it seems to me that the ultimate impact of the present work lies in how possible it will be to uncover the relevant genetic changes underlying the novelty. I don't have any specific experiments to suggest (and I'd be reluctant to suggest major new experiments anyway), but the uncertainty about making progress does constrain my enthusiasm somewhat. And in the meantime, the title ("Expansion of aECM underlies the morphogenesis.…") is a bit misleading and should reflect more this uncertainty.

The comments above address several interconnected issues, which will be responded to in sequence below:

1) It is not clear whether changes in Dumpy expression were at all causative in the evolution of the posterior lobe.

We agree that it would be difficult to conclusively state that changes in Dumpy expression (either by *cis* or *trans* changes) were causative for generating the posterior lobe. However, that is not the main conclusion of this paper – our results show how the expansion of an apical ECM is associated with and required for the novel posterior lobe structure. It is important to recognize that the logic of our conclusions follow well-accepted practices in the study of novelty on macroevolutionary scales in the evo-devo field. Specifically, one describes a feature of an evolved trait (e.g. a gene expression pattern), shows how it is unique to the species bearing the trait, and then demonstrates (in the best case scenario) that the feature is required for trait formation. Similar inferences have been made to varying extents in the cases of beetle horns (Moczek and Rose, 2009), treehopper helmets (Fisher et al., 2019; Prud’homme et al., 2011), and butterfly pigment patterns (Kunte et al., 2014; Mazo-Vargas et al., 2017; Reed et al., 2011).

2) It is not entirely surprising that disrupting Dumpy, which attaches to the developing lobe, disrupts its ultimate morphology.

We agree that it is not surprising to see a phenotype once one knows about the vast aECM network associated with the posterior lobe that we uncovered. We would assert that what *is* surprising is the existence of this complex aECM network, which shows signs of evolving connections to cells of the posterior lobe.

3) What is missing is the complementary gain-of-function experiment: does expanding Dumpy expression in a non-lobed species create anything akin to a new lobe?

We would love to test the sufficiency of changes at Dumpy. However, there are three reasons that this would be nearly impossible to perform.

First, we strongly suspect the aECM is composed of multiple proteins in addition to Dumpy. This may include receptors, other ZP proteins, enzymes which modify these proteins and additional components that await discovery as the cell biology of the aECM is largely unexplored. Hence, we agree with the reviewer that the misexpression of just Dumpy is unlikely to drive a substantial response on its own. We discuss this possibility in more detail:

“In addition, while Dumpy may be required for the development of the posterior lobe, additional components of the aECM, including factors that remodel the aECM or receptors that anchor the aECM to the cells, are likely also needed.” (subsection “Integrating cells into a pre-existing aECM network to generate morphological novelty”)

Second, as the reviewer astutely notes, this experiment is technically difficult. We would need validated enhancer-Gal4 drivers to express Dumpy in a non-lobed species. Not just any driver would suffice, but one that has the correct spatiotemporal dynamics to connect lateral plate cells to other parts of the aECM network to mechanically couple these elements together.

Third, generating a UAS construct for dumpy itself is challenging. Dumpy encodes a 2.5 megadalton protein, and its CDNAs are 40-50 kilobases long. Thus generation of a UAS construct for this protein would likely require BAC-scale transgenic manipulations that have not been commonplace in *Drosophila* species outside of *D. melanogaster*, and in fact have not even been performed to our knowledge for *dumpy* in *D. melanogaster*. Given these limitations, we hope that the reviewers and editors can see how this is a massive project that would take several years for any group to accomplish, and may not lead to positive results for good reasons that do not invalidate our conclusions.

4) But then the question remains: how do we solve the "major looming challenge in evo-devo research" that the authors note in their final sentence (Discussion section) -- what are the genetic changes underlying the cellular changes? Moving in that direction would indeed be a major advance, and it seems to me that the ultimate impact of the present work lies in how possible it will be to uncover the relevant genetic changes underlying the novelty.

This is a comment on our final sentence of the discussion, which we wrote for the purpose of highlighting the future of this system. We are therefore excited about the reviewer’s enthusiasm for our stated vision. What we disagree with is the question of whether the work presented here represents a “major advance”. Clearly we think so (for several reasons):

- The posterior lobe is the only trait that can physically distinguish the most extensively studied multicellular species on the planet from its close relatives, and yet the cellular basis of its formation had not been studied.

- We find that the lobe is a single cell tall. This was unexpected to us and colleagues that we have discussed our findings with.

- The molecular basis of extreme apico-basal cell elongation is poorly understood.

- The apical ECM is also a poorly understood participant in epithelial morphogenesis, and has not been thus far implicated in morphological evolution.

- The genital aECM that we describe is unique among apical ECMs in its complexity (connecting to multiple structures), and for its previously undescribed role in extreme apico-basal elongation of epithelial cells.

5) And in the meantime, the title ("Expansion of aECM underlies the morphogenesis.…") is a bit misleading and should reflect more this uncertainty.

We have revised the title to better reflect our conclusions:

”Evolutionary expansion of apical extracellular matrix is required for the elongation of cells in a novel structure”